# A Simple and General Graph Neural Network with Stochastic Message Passing

## Abstract

Graph neural networks (GNNs) are emerging machine learning models on graphs. One key property behind the expressiveness of existing GNNs is that the learned node representations are permutation-equivariant. Though being a desirable property for certain tasks, however, permutation-equivariance prevents GNNs from being proximity-aware, i.e., preserving the walk-based proximities between pairs of nodes, which is another critical property for graph analytical tasks. On the other hand, some variants of GNNs are proposed to preserve node proximities, but they fail to maintain permutation-equivariance. How to empower GNNs to be proximity-aware while maintaining permutation-equivariance remains an open problem. In this paper, we propose Stochastic Message Passing (SMP), a general and simple GNN to maintain both proximity-awareness and permutation-equivariance properties. Specifically, we augment the existing GNNs with stochastic node representations learned to preserve node proximities. Though seemingly simple, we prove that such a mechanism can enable GNNs to preserve node proximities in theory while maintaining permutation-equivariance with certain parametrization. Extensive experimental results demonstrate the effectiveness and efficiency of SMP for tasks including node classification and link prediction.

## 1 Introduction

Graph neural networks (GNNs), as generalizations of neural networks in analyzing graphs, have attracted considerable research attention. GNNs have been widely applied to various applications such as social recommendation (Ma et al., 2019), physical simulation (Kipf et al., 2018), and protein interaction prediction (Zitnik & Leskovec, 2017).

One key property of most existing GNNs is permutation-equivariance, i.e., if we randomly permutate the IDs of nodes while maintaining the graph structure, the representations of nodes in GNNs are permutated accordingly. Mathematically, permutation-equivariance reflects one basic symmetric group of graph structures. Although it is a desirable property for tasks such as node or graph classification (Keriven & Peyré, 2019; Maron et al., 2019b), permutation-equivariance also prevents GNNs from being proximity-aware, i.e., permutation-equivariant GNNs cannot preserve walk-based proximities between nodes such as the shortest distance or high-order proximities (see Theorem 1).

Pairwise proximities between nodes are crucial for graph analytical tasks such as link prediction (Hu et al., 2020; You et al., 2019). To enable a proximity-aware GNN, Position-aware GNN (P-GNN) (You et al., 2019)[1] proposes a sophisticated GNN architecture and shows better performance for proximity-aware tasks. But P-GNN needs to explicitly calculate the shortest distance between nodes and its computational complexity is unaffordable for large graphs. Moreover, P-GNN completely ignores the permutation-equivariance property. Therefore, it cannot produce satisfactory results when permutation-equivariance is helpful.

In real-world scenarios, both proximity-awareness and permutation-equivariance are indispensable properties for GNNs. Firstly, different tasks may require different properties. For example, recommendation applications usually require the model to be proximity-aware (Konstas et al., 2009) while permutation-equivariance is a basic assumption in centrality measurements (Borgatti, 2005). Even

---

[1]In (You et al., 2019), the authors consider the special case of shortest distance between nodes and name such property as "position-aware". In this paper, we consider a more general case of any walk-based proximity.

for the same task, different datasets may have different requirements on these two properties. Taking link prediction as an example, we observe that permutation-equivariant GNNs such as GCN (Kipf & Welling, 2017) or GAT (Velickovic et al., 2018) show better results than P-GNN in coauthor graphs, but the opposite in biological graphs (please see Section 5.2 for details). Unfortunately, in the current GNN frameworks, these two properties are contradicting, as we show in Theorem 1. Whether there exists a general GNN to be proximity-aware while maintaining permutation-equivariance remains an open problem.

In this paper, we propose Stochastic Message Passing (SMP), a general and simple GNN to preserve both proximity-awareness and permutation-equivariance properties. Specifically, we augment the existing GNNs with stochastic node representations learned to preserve proximities. Though seemingly simple, we prove that our proposed SMP can enable GNNs to preserve walk-based proximities in theory (see Theorem 2 and Theorem 3). Meanwhile, SMP is equivalent to a permutation-equivariant GNN with certain parametrization and thus is at least as powerful as those GNNs in permutation-equivariant tasks (see Remark 1). Therefore, SMP is general and flexible in handling both proximity-aware and permutation-equivariant tasks, which is also demonstrated by our extensive experimental results. Besides, owing to the simple structure, SMP is computationally efficient, with a running time roughly the same as those of the most simple GNNs such as SGC (Wu et al., 2019) and is at least an order of magnitude faster than P-GNN on large graphs. Ablation studies further show that a linear instantiation of SMP is expressive enough as adding extra non-linearities does not lift the performance of SMP on the majority of datasets. Our contributions are as follows.

- We propose SMP, a simple and general GNN to handle both proximity-aware and permutation-equivariant graph analytical tasks.
- We prove that SMP has theoretical guarantees in preserving walk-based proximities and is at least as powerful as the existing GNNs in permutation-equivariant tasks.
- Extensive experimental results demonstrate the effectiveness and efficiency of SMP. We show that a linear instantiation of SMP is expressive enough on the majority of datasets.

## 2 RELATED WORK

We briefly review GNNs and their permutation-equivariance and proximity-awareness property.

The earliest GNNs adopt a recursive definition of node states (Scarselli et al., 2008; Gori et al., 2005) or a contextual realization (Micheli, 2009). GGS-NNs (Li et al., 2016) replace the recursive definition with recurrent neural networks (RNNs). Spectral GCNs (Bruna et al., 2014) defined graph convolutions using graph signal processing (Shuman et al., 2013; Ortega et al., 2018) with Cheb-Net (Defferrard et al., 2016) and GCN (Kipf & Welling, 2017) approximating the spectral filters using a $K$-order Chebyshev polynomial and the first-order polynomial, respectively. MPNNs (Gilmer et al., 2017), GraphSAGE (Hamilton et al., 2017), and MoNet (Monti et al., 2017) are proposed as general frameworks by characterizing GNNs with a message-passing function and an updating function. More advanced variants such as GAT (Velickovic et al., 2018), JK-Nets (Xu et al., 2018b), GIN (Xu et al., 2018a), and GraphNets (Battaglia et al., 2018) follow these frameworks.

Li *et al.* (Li et al., 2018), Xu *et al.* (Xu et al., 2018a), Morris *et al.* (Morris et al., 2019), and Maron *et al.* (Maron et al., 2019a) show the connection between GNNs and the Weisfeiler-Lehman algorithm (Shervashidze et al., 2011) of graph isomorphism tests, in which permutation-equivariance holds a key constraint. Maron *et al.* (Maron et al., 2019b) and Keriven *et al.* (Keriven & Peyré, 2019) analyze the permutation-equivariance property of GNNs more theoretically. To date, most of the existing GNNs are permutation-equivariant and thus are not proximity-aware. The only exception is P-GNN (You et al., 2019), which proposes to capture the positions of nodes using the relative distance between the target node and some randomly chosen anchor nodes. However, P-GNN cannot satisfy permutation-equivariance and is computationally expensive.

Very recently, motivated by enhancing the expressive power of GNNs in graph isomorphism tests and distributed computing literature (Angluin, 1980; Linial, 1992; Naor & Stockmeyer, 1995), some studies suggest assigning unique node identifiers for GNNs (Loukas, 2020) such as one-hot IDs (Murphy et al., 2019) or random numbers (Dasoulas et al., 2019; Sato et al., 2020; Corso et al., 2020). For example, Sato *et al.* (Sato et al., 2020) novelly show that random numbers can enhance GNNs in tackling two important graph-based NP problems with a theoretical guarantee, namely the

minimum dominating set and the maximum matching problem, and Fey *et al.* (Fey et al., 2020) empirically show the effectiveness of random features in the graph matching problem. Our work differs in that we systematically study how to preserve permutation-equivariance and proximity-awareness simultaneously in a simple yet effective framework, which is a new topic different from these existing works. Besides, we theoretically prove that our proposed method can preserve walk-based proximities by using the random projection literature. We also demonstrate the effectiveness of our method on various large-scale benchmarks for both node- and edge-level tasks, while no similar results are reported in the literature.

The design of our method is also inspired by the random projection literature in dimensionality reduction (Vempala, 2005) and to the best of our knowledge, we are the first to study random projection in the scope of GNNs. More remotely, our definition of node proximities is inspired and inherited from graph kernels (Gärtner et al., 2003; Borgwardt & Kriegel, 2005), network embedding (Perozzi et al., 2014; Grover & Leskovec, 2016), and general studies of graphs (Newman, 2018).

## 3 MESSAGE-PASSING GNNs

We consider a graph $G = (\mathcal{V}, \mathcal{E}, \mathbf{F})$ where $\mathcal{V} = \{v_1, ..., v_N\}$ is the set of $N = |\mathcal{V}|$ nodes, $\mathcal{E} \subseteq \mathcal{V} \times \mathcal{V}$ is the set of $M = |\mathcal{E}|$ edges, and $\mathbf{F} \in \mathbb{R}^{N \times d_0}$ is a matrix of $d_0$ node features. The adjacency matrix is denoted as $\mathbf{A}$, where its $i^{th}$ row, $j^{th}$ column and an element denoted as $\mathbf{A}_{i,:}$, $\mathbf{A}_{:,j}$, and $\mathbf{A}_{i,j}$, respectively. In this paper, we assume the graph is unweighted and undirected. The neighborhood of node $v_i$ is denoted as $\mathcal{N}_i$ and $\tilde{\mathcal{N}}_i = \mathcal{N}_i \cup \{v_i\}$.

The existing GNNs usually follow a message-passing framework (Gilmer et al., 2017), where the $l^{th}$ layer adopts a neighborhood aggregation function AGG($\cdot$) and an updating function UPDATE($\cdot$):

$$\mathbf{m}_i^{(l)} = \text{AGG}(\{\mathbf{h}_j^{(l)}, \forall j \in \tilde{\mathcal{N}}_i\}), \mathbf{h}_i^{(l+1)} = \text{UPDATE}([\mathbf{h}_i^{(l)}, \mathbf{m}_i^{(l)}]), \tag{1}$$

where $\mathbf{h}_i^{(l)} \in \mathbb{R}^{d_l}$ is the representation of node $v_i$ in the $l^{th}$ layer, $d_l$ is the dimensionality, and $\mathbf{m}_i^{(l)}$ are the messages. We also denote $\mathbf{H}^{(l)} = [\mathbf{h}_1^{(l)}, ..., \mathbf{h}_N^{(l)}]$ and $[\cdot, \cdot]$ is the concatenation operation. The node representations are initialized as node features, i.e., $\mathbf{H}^{(0)} = \mathbf{F}$. We denote a GNN following Eq. (1) with $L$ layers as a parameterized function as follows[2]:

$$\mathbf{H}^{(L)} = \mathcal{F}_{\text{GNN}}(\mathbf{A}, \mathbf{F}; \mathbf{W}), \tag{2}$$

where $\mathbf{H}^{(L)}$ are final node representations learned by the GNN and $\mathbf{W}$ denotes all the parameters.

One key property of the existing GNNs is permutation-equivariance.

**Definition 1** (Permutation-equivariance). *Consider a graph $G = (\mathcal{V}, \mathcal{E}, \mathbf{F})$ and any permutation $\mathcal{P} : \mathcal{V} \rightarrow \mathcal{V}$ so that $G' = (\mathcal{V}, \mathcal{E}', \mathbf{F}')$ has an adjacency matrix $\mathbf{A}' = \mathbf{P}\mathbf{A}\mathbf{P}^T$ and a feature matrix $\mathbf{F}' = \mathbf{P}\mathbf{F}$, where $\mathbf{P} \in \{0, 1\}^{N \times N}$ is the permutation matrix corresponding to $\mathcal{P}$, i.e., $\mathbf{P}_{i,j} = 1$ iff $\mathcal{P}(v_i) = v_j$. A GNN satisfies permutation-equivariance if the node representations are equivariant with respect to $\mathcal{P}$, i.e.,*

$$\mathbf{P}\mathcal{F}_{GNN}(\mathbf{A}, \mathbf{F}; \mathbf{W}) = \mathcal{F}_{GNN}(\mathbf{P}\mathbf{A}\mathbf{P}^T, \mathbf{P}\mathbf{F}; \mathbf{W}). \tag{3}$$

It is known that GNNs following Eq. (1) are permutation-equivariant (Maron et al., 2019b).

**Definition 2** (Automorphism). *A graph $G$ is said to have (non-trivial) automorphism if there exists a non-identity permutation matrix $\mathbf{P} \neq \mathbf{I}_N$ so that $\mathbf{A} = \mathbf{P}\mathbf{A}\mathbf{P}^T$ and $\mathbf{F} = \mathbf{P}\mathbf{F}$. We denote the corresponding automorphic node pairs as $\mathcal{C}_G = \bigcup_{\mathbf{P} \neq \mathbf{I}_N} \{(i, j) | \mathbf{P}_{i,j} \neq 0, i \neq j\}$*

**Corollary 1.** *Using Definition 1 and 2, if a graph has automorphism, a permutation-equivariant GNN will produce identical node representations for automorphic node pairs:*

$$\mathbf{h}_i^{(L)} = \mathbf{h}_j^{(L)}, \forall (i, j) \in \mathcal{C}_G. \tag{4}$$

Since the node representations are used for downstream tasks, the corollary shows that permutation-equivariant GNNs cannot differentiate automorphic node pairs. A direct consequence of Corollary 1 is that permutation-equivariant GNNs cannot preserve walk-based proximities between pairs of nodes. The formal definitions are as follows.

---

[2]Since the final layer of GNNs is task-specific, e.g., a softmax layer for node classification or a readout layer for graph classification, we only consider the GNN architecture to its last hidden layer.

**Definition 3** (Walk-based Proximities). *For a given graph $G = (\mathcal{V}, \mathcal{E}, \mathbf{F})$, we use a matrix $\mathbf{S} \in \mathbb{R}^{N \times N}$ to denote walk-based proximities between pairs of nodes defined as:*

$$\mathbf{S}_{i,j} = \mathcal{S}\left(\{v_i \rightsquigarrow v_j\}\right), \tag{5}$$

*where $v_i \rightsquigarrow v_j$ denotes walks from node $v_i$ to $v_j$ and $\mathcal{S}(\cdot)$ is an arbitrary real-valued function. The length of a walk-based proximity is the maximum length of all the walks of the proximity.*

Typical examples of walk-based proximities include the shortest distance (You et al., 2019), the high-order proximities (a sum of walks weighted by their lengths) (Zhang et al., 2018), and random walk probabilities (Klicpera et al., 2019). Next, we give a definition of preserving walk-based proximities.

**Definition 4.** *For a given walk-based proximity, a GNN is said to be able to preserve the proximity if there exists a decoder function $\mathcal{F}_{de}(\cdot)$ satisfying that for any graph $G = (\mathcal{V}, \mathcal{E}, \mathbf{F})$, there exist parameters $\mathbf{W}_G$ so that $\forall \epsilon > 0$:*

$$\left| \mathbf{S}_{i,j} - \mathcal{F}_{de}\left(\mathbf{H}_{i,:}^{(L)}, \mathbf{H}_{j,:}^{(L)}\right) \right| < \epsilon, \tag{6}$$

*where*

$$\mathbf{H}^{(L)} = \mathcal{F}_{GNN}(\mathbf{A}, \mathbf{F}; \mathbf{W}_G). \tag{7}$$

Note that we do not constrain the GNN architecture as long as it follows Eq. (1), and the decoder function is also arbitrary (but notice that it cannot take the graph structure as inputs). In fact, both the GNN and the decoder function can be arbitrarily deep and with sufficient hidden units.

**Theorem 1.** *The existing permutation-equivariant GNNs cannot preserve any walk-based proximity except the trivial solution that all node pairs have the same proximity.[3]*

The formulation and proof of the theorem are given in Appendix A.1. Since walk-based proximities are rather general and widely adopted in graph analytical tasks such as link prediction, the theorem shows that the existing permutation-equivariant GNNs cannot handle these tasks well.

## 4 THE MODEL

### 4.1 A GNN FRAMEWORK USING STOCHASTIC MESSAGE PASSING

A major shortcoming of permutation-equivariant GNNs is that they cannot differentiate automorphic node pairs. To solve that problem, we need to introduce some mechanism as "symmetry breaking", i.e., to enable GNNs to distinguish these nodes. To achieve this goal, we sample a stochastic matrix $\mathbf{E} \in \mathbb{R}^{N \times d}$ where each element follows an i.i.d. normal distribution $\mathcal{N}(0, 1)$. The stochastic matrix can provide signals in distinguishing the nodes because they are randomly sampled without being affected by the graph automorphism. In fact, we can easily calculate that the Euclidean distance between two stochastic signals divided by a constant $\sqrt{2}$ follows a chi distribution $\chi_d$:

$$\frac{1}{\sqrt{2}} \left| \mathbf{E}_{i,:} - \mathbf{E}_{j,:} \right| \sim \chi_d, \forall i, j. \tag{8}$$

When $d$ is reasonably large, e.g., $d > 20$, the probability of two signals being close is very low. Then, inspired by the message-passing framework, we apply a GNN on the stochastic matrix so that nodes can exchange information of the stochastic signals:

$$\tilde{\mathbf{E}} = \mathcal{F}_{GNN}\left(\mathbf{A}, \mathbf{E}; \mathbf{W}\right). \tag{9}$$

We call $\tilde{\mathbf{E}}$ the stochastic representation of nodes. Using the stochastic matrix and message-passing, $\tilde{\mathbf{E}}$ can be used to preserve node proximities (see Theorem 2 and Theorem 3). Then, to let our model still be able to utilize node features, we concatenate $\tilde{\mathbf{E}}$ with the node representations from another GNN with node features as inputs:

$$\mathbf{H} = \mathcal{F}_{\text{output}}([\tilde{\mathbf{E}}, \mathbf{H}^{(L)}])$$
$$\tilde{\mathbf{E}} = \mathcal{F}_{GNN}\left(\mathbf{A}, \mathbf{E}; \mathbf{W}\right), \mathbf{H}^{(L)} = \mathcal{F}_{GNN'}(\mathbf{A}, \mathbf{F}; \mathbf{W}'), \tag{10}$$

---

[3]Proposition 1 in (You et al., 2019) can be regarded as a special case of Theorem 1 using the shortest distance proximity.

where $\mathcal{F}_{\text{output}}(\cdot)$ is an aggregation function such as a linear function or simply the identity mapping. In a nutshell, our proposed method augments the existing GNNs with a stochastic representation learned by message-passings to differentiate different nodes and preserve node proximities.

There is also a delicate choice worthy mentioning, i.e., whether the stochastic matrix $\mathbf{E}$ is fixed or resampled in each epoch. By fixing $\mathbf{E}$, the model can learn to memorize the stochastic representation and distinguish different nodes, but with the cost of unable to handle nodes not seen during training. On the other hand, by resampling $\mathbf{E}$ in each epoch, the model can have a better generalization ability since the model cannot simply remember one specific stochastic matrix. However, the node representations are not fixed (but pairwise proximities are preserved; see Theorem 2). In these cases, $\tilde{\mathbf{E}}$ is more capable of handling pairwise tasks such as link prediction or pairwise node classification. In this paper, we use a fixed $\tilde{\mathbf{E}}$ for transductive datasets and resample $\mathbf{E}$ for inductive datasets.

**Time Complexity** From Eq.(10), the time complexity of our framework mainly depends on the two GNNs in learning the stochastic and permutation-equivariant node representations. In this paper, we instantiate these two GNNs using simple message-passing GNNs such as GCN (Kipf & Welling, 2017) and SGC (Wu et al., 2019) (see Section 4.2 and Section 4.3). Thus, the time complexity of our method is the same as these models, which is $O(M)$, i.e., linear with respect to the number of edges. We also empirically compare the running time of different models in Appendix 5.5. Besides, many acceleration schemes for GNNs such as sampling (Chen et al., 2018a;b; Huang et al., 2018) or partitioning the graph (Chiang et al., 2019) can be directly applied to our framework.

## 4.2 A LINEAR INSTANTIATION

Based on the general framework shown in Eq. (10), we attempt to explore its minimum model instantiation, i.e., a linear model. Specifically, inspired by Simplified Graph Convolution (SGC) (Wu et al., 2019), we adopt a linear message-passing for both GNNs, i.e.,

$$\mathbf{H} = \mathcal{F}_{\text{output}}([\tilde{\mathbf{E}}, \mathbf{H}^{(L)}]) = \mathcal{F}_{\text{output}}\left(\left[\tilde{\mathbf{A}}^K \mathbf{E}, \tilde{\mathbf{A}}^K \mathbf{F}\right]\right), \tag{11}$$

where $\tilde{\mathbf{A}} = (\mathbf{D}+\mathbf{I})^{-\frac{1}{2}}(\mathbf{A}+\mathbf{I})(\mathbf{D}+\mathbf{I})^{-\frac{1}{2}}$ is the normalized graph adjacency matrix with self-loops proposed in GCN (Kipf & Welling, 2017) and $K$ is the number of propagation steps. We also set $\mathcal{F}_{\text{output}}(\cdot)$ in Eq. (11) as a linear mapping or identity mapping.

Though seemingly simple, we show that such an SMP instantiation possesses a theoretical guarantee in preserving the walk-based proximities.

**Theorem 2.** *An SMP in Eq.* (11) *with the message-passing matrix $\tilde{\mathbf{A}}$ and the number of propagation steps $K$ can preserve the walk-based proximity $\tilde{\mathbf{A}}^K(\tilde{\mathbf{A}}^K)^T$ with high probability if the dimensionality of the stochastic matrix $d$ is sufficiently large, where the superscript $T$ denotes matrix transpose. The theorem is regardless of whether $\mathbf{E}$ are fixed or resampled.*

The mathematical formulation and proof of the theorem are given in Appendix A.2. In addition, we show that SMP is equivalent to a permutation-equivariant GNN with certain parametrization.

**Remark 1.** *Suppose we adopt $\mathcal{F}_{output}(\cdot)$ as a linear function with the output dimensionality the same as $\mathcal{F}_{GNN'}$. Then, Eq.* (10) *is equivalent to the permutation-equivariant $\mathcal{F}_{GNN'}(\mathbf{A}, \mathbf{F}; \mathbf{W}')$ if the parameters in $\mathcal{F}_{output}(\cdot)$ are all-zeros for $\tilde{E}$ and an identity matrix for $\mathbf{H}^{(L)}$.*

The result is straightforward from the definition. Then, we have the following corollary.

**Corollary 2.** *For any task, Eq.* (10) *with the aforementioned linear $\mathcal{F}_{output}(\cdot)$ is at least as powerful as the permutation-equivariant $\mathcal{F}_{GNN'}(\mathbf{A}, \mathbf{F}; \mathbf{W}')$, i.e., the minimum training loss of using $\mathbf{H}$ in Eq.* (10) *is equal to or smaller than using $\mathbf{H}^{(L)} = \mathcal{F}_{GNN'}(\mathbf{A}, \mathbf{F}; \mathbf{W}')$.*

In other words, SMP will not hinder the performance[4] even the tasks are permutation-equivariant since the stochastic representations are concatenated with the permutation-equivariant GNNs followed by a linear mapping. In these cases, the linear SMP is equivalent to SGC (Wu et al., 2019).

Combining Theorem 2 and Corollary 2, the linear SMP instantiation in Eq. (11) is capable of handling both proximity-aware and permutation-equivariant tasks.

---

[4]Similar to previous works such as (Hamilton et al., 2017; Xu et al., 2018a), we only consider the minimum training loss because the optimization landscapes and generalization gaps are difficult to analyze analytically.

### 4.3 NON-LINEAR EXTENSIONS

One may question whether a more sophisticated variant of Eq. (10) can further improve the expressiveness of SMP. There are three adjustable components in Eq. (10): two GNNs in propagating the stochastic matrix and node features, respectively, and an output function. In theory, adopting non-linear models as either component is able to enhance the expressiveness of SMP. Indeed, if we use a sufficiently expressive GNN in learning $\tilde{\mathbf{E}}$ instead of linear propagations, we can prove a more general version of Theorem 2 as follows.

**Theorem 3.** *An SMP variant following Eq.(10) with $\mathcal{F}_{GNN}(\mathbf{A}, \mathbf{E}; \mathbf{W})$ containing $L$ layers can preserve any length-$L$ walk-based proximity if the message-passing and updating functions in the GNN are sufficiently expressive. In this theorem, we also assume the Gaussian random vectors $\mathbf{E}$ are rounded to machine precision so that $\mathbf{E}$ is drawn from a countable subspace of $\mathbb{R}$.*

The proof of the theorem is given in Appendix A.3. Similarly, we can adopt more advanced methods for $\mathcal{F}_{\text{output}}(\cdot)$ such as gating or attention so that the two GNNs are more properly integrated.

Although non-linear extensions of SMP can, in theory, increase the model expressiveness, they also take a higher risk of over-fitting due to model complexity, not to mention that the computational cost will also increase. In practice, we find in ablation studies that the linear SMP instantiation in Eq. (11) works reasonably well on most of the datasets (please refer to Section 5.4 for further details).

## 5 EXPERIMENTS

### 5.1 EXPERIMENTAL SETUPS

**Datasets** We conduct experiments on the following **ten** datasets: two simulation datasets, **Grid** and **Communities** (You et al., 2019), a communication dataset **Email** (You et al., 2019), two coauthor networks, **CS** and **Physics** (Shchur et al., 2018), two protein interaction networks, **PPI** (Hamilton et al., 2017) and **PPA** (Hu et al., 2020), and three GNN benchmarks, **Cora**, **CiteSeer**, and **PubMed** (Yang et al., 2016). We only report the results of three benchmarks for the node classification task and the results for other tasks are shown in Appendix B due to the page limit. More details of the datasets including their statistics are provided in Appendix C.1. These datasets cover a wide spectrum of domains, sizes, and with or without node features. Since Email and PPI contain more than one graph, we conduct experiments in an *inductive setting* on these two datasets, i.e., the training, validation, and testing set are split with respect to different graphs.

**Baselines** We adopt two sets of baselines. The first set is permutation-equivariant GNNs including GCN (Kipf & Welling, 2017), GAT (Velickovic et al., 2018), and SGC (Wu et al., 2019), which are widely adopted GNN architectures. The second set contains P-GNN (You et al., 2019), the only proximity-aware GNN to date. We use the P-GNN-F version.

In comparing with the baselines, we mainly evaluate two variants of SMP with different $\mathcal{F}_{\text{output}}(\cdot)$: SMP-Identity, i.e., $\mathcal{F}_{\text{output}}(\cdot)$ as an identity mapping, and SMP-Linear, i.e., $\mathcal{F}_{\text{output}}(\cdot)$ as a linear mapping. Note that both variants adopt linear message-passing functions as SGC. We conduct more ablation studies with different SMP variants in Section 5.4.

For fair comparisons, we adopt the same architecture and hyper-parameters for all the methods (please refer to Appendix C.2 for the details). For datasets without node features, we adopt a constant vector as the node features. We experiment on two tasks: link prediction and node classification. Additional experiments on graph reconstruction, pairwise node classification, and running time comparison are provided in Appendix B. We repeat the experiments 10 times for datasets except for PPA and 3 times for PPA, and report the average results.

### 5.2 LINK PREDICTION

Link prediction aims to predict missing links of a graph. Specifically, we split the edges into 80%-10%-10% and use them for training, validation, and testing, respectively. Besides adopting those real edges as positive samples, we obtain negative samples by randomly sampling an equal number of node pairs that do not have edges. For all the methods, we set a simple classifier: $\text{Sigmoid}(\mathbf{H}_i^T \mathbf{H}_j)$, i.e., use the inner product to predict whether a node pair $(v_i, v_j)$ forms a link, and use AUC (area

Table 2: The results of link prediction tasks measured in AUC (%). The best results and the second-best results for each dataset, respectively, are in bold and underlined.

| Model | Grid | Communities | Email | CS | Physics | PPI |
|---|---|---|---|---|---|---|
| SGC | 57.6±3.8 | 51.9±1.6 | 68.5±7.0 | 96.5±0.1 | **96.6±0.1** | 80.5±0.4 |
| GCN | 61.8±3.6 | 50.3±2.5 | 67.4±6.9 | 93.4±0.3 | 93.8±0.2 | 78.0±0.4 |
| GAT | 61.0±5.5 | 51.1±1.6 | 53.5±6.3 | 93.7±0.9 | 94.1±0.4 | 79.3±0.5 |
| PGNN[5] | 73.4±6.0 | 97.8±0.6 | 70.9±6.4 | 82.2±0.5 | Out of memory | 80.8±0.4 |
| SMP-Identity | 55.1±4.8 | **98.0±0.7** | 72.9±5.1 | 96.5±0.1 | 96.5±0.1 | 81.0±0.2 |
| SMP-Linear | **73.6±6.2** | 97.7±0.5 | **75.7±5.0** | **96.7±0.1** | 96.1±0.1 | **81.9±0.3** |

under the curve) as the evaluation metric. One exception to the aforementioned setting is that on the PPA dataset, we follow the splits and evaluation metric (i.e., Hits@100) provided by the dataset (Hu et al., 2020). The results except PPA are shown in Table 2. We make the following observations.

- Our proposed SMP achieves the best results on five out of the six datasets and is highly competitive (the second-best result) on the other (Physics). The results demonstrate the effectiveness of our proposed method on link prediction tasks. We attribute the strong performance of SMP to its capability of maintaining both proximity-awareness and permutation-equivariance properties.

- On Grid, Communities, Email, and PPI, both SMP and P-GNN outperform the permutation-equivariant GNNs, proving the importance of preserving node proximities. Although SMP is simpler and more computationally efficient than P-GNN, SMP reports even better results.

- When node features are available (CS, Physics, and PPI), SGC can outperform GCN and GAT. The results re-validate the experiments in SGC (Wu et al., 2019) that the non-linearity in GNNs is not necessarily indispensable. Some plausible reasons include that the additional model complexity brought by non-linear operators makes the models tend to overfit and also difficult to train (see Appendix B.6). On those datasets, SMP retains comparable performance on two coauthor graphs and shows better performance on PPI, possibly because node features on protein graphs are less informative than node features on coauthor graphs for predicting links, and thus preserving graph structure is more beneficial on PPI.

- As Email and PPI are conducted in an inductive setting, i.e., using different graphs for training/validation/testing, the results show that SMP can handle inductive tasks as well.

The results on PPA are shown in Table 1. SMP again outperforms all the baselines, showing that it can handle large-scale graphs with millions of nodes and edges. PPA is part of a recently released Open Graph Benchmark (Hu et al., 2020). The superior performance on PPA further demonstrates the effectiveness of our proposed method in the link prediction task.

Table 1: The results of link prediction on the PPA dataset. The best result and the second-best result are in bold and underlined, respectively.

| Model | Hits@100 |
|---|---|
| SGC | 0.1187±0.0012 |
| GCN | 0.1867±0.0132 |
| GraphSAGE | 0.1655±0.0240 |
| P-GNN | Out of Memory |
| Node2vec | 0.2226±0.0083 |
| Matrix Factorization | 0.3229±0.0094 |
| SMP-Identity | 0.2018±0.0148 |
| SMP-Linear | **0.3582±0.0070** |

### 5.3 NODE CLASSIFICATION

Next, we conduct experiments of node classification, i.e., predicting the labels of nodes. Since we need ground-truths in the evaluation, we only adopt datasets with node labels. Specifically, for CS and Physics, following (Shchur et al., 2018), we adopt 20/30 labeled nodes per class for training/validation and the rest for testing. For Communities, we adjust the number as 5/5/10 labeled nodes per class for training/validation/testing. For Cora, CiteSeer, and PubMed, we use the default splits that came with the datasets. We do not adopt Email because some graphs in the dataset are too small to show stable results and exclude PPI as it is a multi-label dataset.

---

[5]The results of PGNN are slightly different compared to the paper because we adopt a more practical and common setting that negative samples in the data are not known apriori but randomly sampled in each epoch.

We use a softmax layer on the learned node representations as the classifier and adopt accuracy, i.e., how many percentages of nodes are correctly classified, as the evaluation criteria. We omit the results of SMP-Identity for this task since the node representations in SMP-Identity have a fixed dimensionality that does not match the number of classes.

Table 3: The results of node classification tasks measured by accuracy (%). The best results and the second-best results for each dataset, respectively, are in bold and underlined.

| Model | Communities | CS | Physics | Cora | CiteSeer | PubMed |
|---|---|---|---|---|---|---|
| SGC | 7.1±2.1 | 67.2±12.8 | 92.3±1.6 | 76.9±0.2 | 63.6±0.0 | 74.2±0.1 |
| GCN | 7.5±1.2 | 91.1±0.7 | 93.1±0.8 | 81.4±0.5 | **71.3±0.5** | **79.3±0.4** |
| GAT | 5.0±0.0 | 90.5±0.5 | **93.1±0.4** | **82.9±0.5** | 71.2±0.6 | 77.9±0.5 |
| PGNN | 5.2±0.5 | 77.6±7.6 | Out of memory | 59.2±1.5 | 55.7±0.9 | Out of memory |
| SMP-Linear | **99.9±0.3** | **91.5±0.8** | 93.1±0.8 | 80.9±0.8 | 68.2±1.0 | 76.5±0.8 |

The results are shown in Table 3. From the table, we observe that SMP reports nearly perfect results on Communities. Since the node labels are generated by graph structures on Communities and there are no node features, the model needs to be proximity-aware to handle it well. P-GNN, which shows promising results in the link prediction task, also fails miserably here.

On the other five graphs, SMP reports highly competitive performance. These graphs are commonly-used benchmarks for GNNs. P-GNN, which completely ignores permutation-equivariance, performs poorly as expected. In contrast, SMP can manage to recover the permutation-equivariant GNNs and avoid being misled, as proven in Remark 1. In fact, SMP even shows better results than its counterpart, SGC, indicating that preserving proximities is also helpful for these datasets.

## 5.4 ABLATION STUDIES

We conduct ablation studies by comparing different SMP variants, including SMP-Identity, SMP-Linear, and the additional three variants as follows:

- SMP-MLP: we set $\mathcal{F}_{\text{output}}(\cdot)$ as a fully-connected network with 1 hidden layer.
- SMP-Linear-GCN$_{\text{feat}}$: we set $\mathcal{F}_{\text{GNN}'}(\mathbf{A}, \mathbf{F}; \mathbf{W}')$ in Eq. (10) to be a GCN (Kipf & Welling, 2017), i.e., induce non-linearity in message passing for features. $\mathcal{F}_{\text{output}}(\cdot)$ is still linear.
- SMP-Linear-GCN$_{\text{both}}$: we set both $\mathcal{F}_{\text{GNN}}(\mathbf{A}, \mathbf{E}; \mathbf{W})$ and $\mathcal{F}_{\text{GNN}'}(\mathbf{A}, \mathbf{F}; \mathbf{W}')$ to be a GCN (Kipf & Welling, 2017), i.e., induce non-linearity in message passing for both features and stochastic representations. $\mathcal{F}_{\text{output}}(\cdot)$ is linear.

We show the results for link prediction tasks in Table 4. The results for node classification and pairwise node classification, which imply similar conclusions, are provided in Table 10 and Table 11 in Appendix B.5. We make the following observations.

- In general, SMP-Linear shows good-enough performance, achieving the best or second-best results on six datasets and highly competitive on the other (Communities). SMP-Identity, which does not have parameters in the output function, performs slightly worse. The results demonstrate the importance of adopting a learnable linear layer in the output function, which is consistent with Remark 1. SMP-MLP does not lift the performance in general, showing that adding extra complexities in $\mathcal{F}_{\text{output}}(\cdot)$ brings no gain in those datasets.
- SMP-Linear-GCN$_{\text{feat}}$ reports the best results on Communities, PPI, and PPA, indicating that adding extra non-linearities in propagating node features are helpful for some graphs.
- SMP-Linear-GCN$_{\text{both}}$ reports the best results on Gird with a considerable margin. Recall that Grid has no node features. The results indicate that inducing non-linearities can help the stochastic representations capture more proximities, which is more helpful for featureless graphs.

## 5.5 EFFICIENCY COMPARISON

To compare the efficiency of different methods quantitatively, we report the running time of different methods in Table 5. The results are averaged over 3,000 epochs on an NVIDIA TESLA M40 GPU

Table 4: The ablation study of different SMP variants for the link prediction task. Datasets except PPA are measured by AUC (%) and PPA is measured by Hits@100. The best results and the second-best results for each dataset are in bold and underlined, respectively.

| Model | Grid | Communities | Email | CS | Physics | PPI | PPA |
|---|---|---|---|---|---|---|---|
| SMP-Identity | 55.1±4.8 | **98.0±0.7** | 72.9±5.1 | 96.5±0.1 | **96.5±0.1** | 81.0±0.2 | 0.2018±0.0148 |
| SMP-Linear | 73.6±6.2 | 97.7±0.5 | **75.7±5.0** | **96.7±0.1** | 96.1±0.1 | 81.9±0.3 | 0.3582±0.0070 |
| SMP-MLP | 72.1±4.3 | 97.8±0.6 | 62.7±8.1 | 88.9±0.8 | 89.2±0.4 | 80.1±0.3 | 0.2035±0.0038 |
| SMP-Linear-GCN$_{feat}$ | 72.8±4.2 | **98.0±0.4** | 74.2±3.9 | 92.9±0.6 | 94.3±0.2 | **82.3±1.0** | **0.4090±0.0087** |
| SMP-Linear-GCN$_{both}$ | **80.5±3.9** | 97.3±0.7 | 73.4±5.5 | 89.8±2.0 | 91.7±0.2 | 79.7±0.3 | 0.2125±0.0232 |

Table 5: The average running time (in milliseconds) for each epoch (including both training and testing), on link prediction task.

| Model | Grid | Communities | Email | CS | Physics | PPI |
|---|---|---|---|---|---|---|
| SGC | 25 | 28 | 58 | 210 | 651 | 704 |
| GCN | 25 | 35 | 75 | 214 | 612 | 784 |
| GAT | 36 | 43 | 140 | 258 | 801 | 919 |
| PGNN | 81 | 84 | 206 | 19,340 | Out of Memory | 6,521 |
| SMP-Identity | 26 | 37 | 96 | 284 | 751 | 840 |
| SMP-Linear | 28 | 26 | 84 | 212 | 616 | 832 |
| SMP-MLP | 23 | 28 | 83 | 237 | 614 | 831 |
| SMP-Linear-GCN$_{feat}$ | 23 | 29 | 90 | 231 | 636 | 855 |
| SMP-Linear-GCN$_{both}$ | 34 | 40 | 95 | 228 | 626 | 895 |

with 12 GB of memory. The results show that SMP is computationally efficient, i.e., only marginally slower than SGC and comparable to GCN. P-GNN is at least an order of magnitude slower except for the extremely small graphs such as Grid, Communities, or Email with no more than a thousand nodes. In addition, the expensive memory cost makes P-GNN unable to work on large-scale graphs.

## 5.6 MORE EXPERIMENTAL RESULTS

Besides the aforementioned experiments, we also conduct experiments on the following tasks: graph reconstruction (Appendix B.1), pairwise node classification (Appendix B.2), and comparing with one-hot IDs (Appendix B.3). Please refer to the Appendix for experimental results and corresponding analyses.

## 6 CONCLUSION

In this paper, we propose SMP, a general and simple GNN to maintain both proximity-awareness and permutation-equivariance properties. We propose to augment the existing GNNs with stochastic node representations learned to preserve node proximities. We prove that SMP can enable GNN to preserve node proximities in theory and is equivalent to a permutation-equivariant GNN with certain parametrization. Experimental results demonstrate the effectiveness and efficiency of SMP. Ablation studies show that a linear SMP instantiation works reasonably well on most of the datasets.

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

## A    THEOREMS AND PROOFS

### A.1    THEOREM 1

Here we formulate and prove Theorem 1.

**Theorem 1.** *For any walk-based proximity function $\mathcal{S}(\cdot)$, a permutation-equivariant GNN cannot preserve $\mathcal{S}(\cdot)$, except the trivial solution that all node pairs have the same proximity, i.e., $\mathbf{S}_{i,j} = c, \forall i, j$, where $c$ is a constant.*

*Proof.* We prove the theorem by contradiction. Assume there exists a non-trivial $\mathcal{S}(\cdot)$ which a permutation-equivariant GNN can preserve. Consider any graph $G = (\mathcal{V}, \mathcal{E}, \mathbf{F})$ and denote $N = |\mathcal{V}|$. We can create $G' = (\mathcal{V}', \mathcal{E}', \mathbf{F}')$ with $|\mathcal{V}'| = 2N$ so that:

$$
\mathcal{E}'_{i,j} = \begin{cases} \mathcal{E}_{i,j} & \text{if } i \le N, j \le N \\ \mathcal{E}_{i-N, j-N} & \text{if } i > N, j > N \\ 0 & \text{else} \end{cases}, \quad \mathbf{F}'_{i,:} = \begin{cases} \mathbf{F}_{i,:} & \text{if } i \le N \\ \mathbf{F}_{i-N,:} & \text{if } i > N \end{cases}.
\tag{12}
$$

Basically, we generate two "copies" of the original graph, one indexing from $1$ to $N$, and the other indexing from $N+1$ to $2N$. By assumption, there exists a permutation-equivariant GNN which can preserve $\mathcal{S}(\cdot)$ in $G'$ and we denote the node representations as $\mathbf{H}'^{(L)} = \mathcal{F}_{\text{GNN}}(\mathbf{A}', \mathbf{F}'; \mathbf{W}_{G'})$. It is easy to see that node $v'_i$ and $v'_{i+N}$ in $G'$ form an automorphic node pair. Using Corollary 1, their representations will be identical in any permutation-equivariant GNN, i.e.,

$$
\mathbf{H}'^{(L)}_{i,:} = \mathbf{H}'^{(L)}_{i+N,:}, \forall i \le N.
\tag{13}
$$

Also, note that there exists no walk from the two copies, i.e. $\{v'_i \rightsquigarrow v'_j\} = \{v'_j \rightsquigarrow v'_i\} = \emptyset, \forall i \le N, j > N$. As a result, for $\forall i \le N, j \le N, \forall \epsilon > 0$, we have:

$$
\begin{aligned}
|\mathbf{S}_{i,j} - \mathcal{S}(\emptyset)| &\le \left| \mathbf{S}_{i,j} - \mathcal{F}_{\text{de}}\left( \mathbf{H}'^{(L)}_{i,:}, \mathbf{H}'^{(L)}_{j,:} \right) \right| + \left| \mathcal{S}(\emptyset) - \mathcal{F}_{\text{de}}\left( \mathbf{H}'^{(L)}_{i,:}, \mathbf{H}'^{(L)}_{j,:} \right) \right| \\
&= \left| \mathbf{S}_{i,j} - \mathcal{F}_{\text{de}}\left( \mathbf{H}'^{(L)}_{i,:}, \mathbf{H}'^{(L)}_{j,:} \right) \right| + \left| \mathbf{S}_{i,j+N} - \mathcal{F}_{\text{de}}\left( \mathbf{H}'^{(L)}_{i,:}, \mathbf{H}'^{(L)}_{j+N,:} \right) \right| < 2\epsilon.
\end{aligned}
\tag{14}
$$

We can prove the same for $\forall i > N, j > N$. The equation naturally holds if $i \le N, j > N$ or $i > N, j \le N$ since $\{v'_i \rightsquigarrow v'_j\} = \emptyset$. Combining the results, we have $\forall \epsilon > 0, \forall i, j, |\mathbf{S}_{i,j} - \mathcal{S}(\emptyset)| < 2\epsilon$. Since $\epsilon$ can be arbitrarily small, the equation shows that all node pairs have the same proximity $c = \mathcal{S}(\emptyset)$, which leads to a contraction and finishes our proof. $\qquad\square$

Notice that in our proof, $G'$ can be constructed for any graph, so rather than designing one specific counter-example, we have shown that there always exists an infinite number of counter-examples by constructing automorphisms in the graph.

Some may find that our counter-examples in the above proof will lead to multiple connected components. Next, we give an alternative proof maintaining one connected component (assuming the original graph is connected) under the assumption that the walk-based proximity is of finite length.

*Proof.* Similar to the previous proof, we assume there exists a non-trivial $\mathcal{S}(\cdot)$ which a permutation-equivariant GNN can preserve. Besides, we assume the length of $\mathcal{S}(\cdot)$ is upper bounded by $l_{\max}$, where $l_{\max}$ is any finite number, i.e., $\forall i, j$,

$$
\mathbf{S}_{i,j} = \mathcal{S}\left( \{v_i \rightsquigarrow v_j\} \right) = \mathcal{S}\left( \{v_i \rightsquigarrow v_j | \text{len}(v_i \rightsquigarrow v_j) \le l_{\max}\} \right).
\tag{15}
$$

Then, for a connected graph $G = (\mathcal{V}, \mathcal{E}, \mathbf{F})$, we create $G' = (\mathcal{V}', \mathcal{E}', \mathbf{F}')$ similar to Eq. (12). Specifically, denoting $\tilde{N} = N + l_{\max}$, we let $G'$ have $3\tilde{N}$ nodes so that:

$$\mathcal{E}'_{i,j} = \begin{cases} \mathcal{E}_{i,j} & \text{if } i,j \leq N \\ 1 & \text{if } N \leq i,j \leq \tilde{N}+1, |j-i|=1 \\ \mathcal{E}_{i-\tilde{N},j-\tilde{N}} & \text{if } \tilde{N} < i,j \leq \tilde{N}+N \\ 1 & \text{if } \tilde{N}+N \leq i,j \leq 2\tilde{N}+1, |j-i|=1 \\ \mathcal{E}_{i-2\tilde{N},j-2\tilde{N}} & \text{if } 2\tilde{N} < i,j \leq 2\tilde{N}+N \\ 1 & \text{if } 2\tilde{N}+N \leq i,j, |j-i|=1 \\ 1 & \text{if } i=3\tilde{N}, j=1 \text{ or } j=3\tilde{N}, i=1 \\ 0 & \text{else} \end{cases}, \mathbf{F}'_{i,:} = \begin{cases} \mathbf{F}_{i,:} & \text{if } i \leq N \\ 0 & \text{if } N < i \leq \tilde{N} \\ \mathbf{F}_{i-\tilde{N},:} & \text{if } \tilde{N} < i \leq \tilde{N}+N \\ 0 & \text{if } \tilde{N}+N < i \leq 2\tilde{N} \\ \mathbf{F}_{i-2\tilde{N},:} & \text{if } 2\tilde{N} < i \leq 2\tilde{N}+N \\ 0 & \text{if } 2\tilde{N}+N < i \end{cases}.$$

$$(16)$$

Intuitively, we create three "copies" of $G$ and three "bridges" to connect the copies and thus make $G'$ also connected. It is also easy to see that nodes $v'_i$, $v'_{i+\tilde{N}}$, and $v'_{i+2\tilde{N}}$ all form automorphic node pairs and thus we have:

$$\mathbf{H}'^{(L)}_{i,:} = \mathbf{H}'^{(L)}_{i+\tilde{N},:} = \mathbf{H}'^{(L)}_{i+2\tilde{N},:}, \forall i \leq \tilde{N}. \tag{17}$$

Next, we can see that the nodes in $G'$ are divided into six parts (three copies and three bridges), which we denote as $\mathcal{V}'_1 = \{v_1, ..., v_N\}$, $\mathcal{V}'_2 = \{v_{N+1}, ..., v_{\tilde{N}}\}$, $\mathcal{V}'_3 = \{v_{\tilde{N}+1}, ..., v_{\tilde{N}+N}\}$, $\mathcal{V}'_4 = \{v_{\tilde{N}+N+1}, ..., v_{2\tilde{N}}\}$, $\mathcal{V}'_5 = \{v_{2\tilde{N}+1}, ..., v_{2\tilde{N}+N}\}$, and $\mathcal{V}'_6 = \{v_{2\tilde{N}+N+1}, ..., v_{3\tilde{N}}\}$. Since $\mathcal{V}'_2, \mathcal{V}'_4, \mathcal{V}'_6$ are bridges with length $l_{\max}$, any walk crosses these bridges will have a length large than $l_{\max}$. For example, let us focus on $v_i \in \mathcal{V}'_1$, i.e., $i \leq N$. If $v_j$ is in $\mathcal{V}'_3, \mathcal{V}'_4$, or $\mathcal{V}'_5$ (i.e., $\tilde{N} < j \leq 2\tilde{N}+N$), any walk $v_i \rightsquigarrow v_j$ will either pass the bridge $\mathcal{V}'_2$ or $\mathcal{V}'_6$ and thus has a length larger than $l_{\max}$. As a result, we have:

$$\mathbf{S}_{i,j} = \mathcal{S}(\{v_i \rightsquigarrow v_j\}) = \mathcal{S}(\{v_i \rightsquigarrow v_j | \text{len}(v_i \rightsquigarrow v_j) \leq l_{\max}\}) = \mathcal{S}(\emptyset). \tag{18}$$

If $v_j \in \mathcal{V}'_1$ or $v_j \in \mathcal{V}'_2$, i.e., $j \leq \tilde{N}$, we can use the fact that $v_j$ and $v_{j+\tilde{N}}$ forms an automorphic node pair similar to Eq. (14), i.e., $\forall \epsilon > 0$, we have

$$\left| \mathbf{S}_{i,j} - \mathcal{S}(\emptyset) \right| \leq \left| \mathbf{S}_{i,j} - \mathcal{F}_{\text{de}}\left(\mathbf{H}'^{(L)}_{i,:}, \mathbf{H}'^{(L)}_{j,:}\right) \right| + \left| \mathcal{S}(\emptyset) - \mathcal{F}_{\text{de}}\left(\mathbf{H}'^{(L)}_{i,:}, \mathbf{H}'^{(L)}_{j,:}\right) \right|$$

$$= \left| \mathbf{S}_{i,j} - \mathcal{F}_{\text{de}}\left(\mathbf{H}'^{(L)}_{i,:}, \mathbf{H}'^{(L)}_{j,:}\right) \right| + \left| \mathbf{S}_{i,j+\tilde{N}} - \mathcal{F}_{\text{de}}\left(\mathbf{H}'^{(L)}_{i,:}, \mathbf{H}'^{(L)}_{j+\tilde{N},:}\right) \right| < 2\epsilon. \tag{19}$$

Similarly, if $v_j \in \mathcal{V}'_6$, i.e., $2\tilde{N}+N < j$, we can use the fact that $v_j$ and $v_{j-\tilde{N}}$ forms an automorphic node pair to prove the same inequality. Thus, we prove that if $i \leq N, \forall \epsilon > 0, \forall j, |\mathbf{S}_{i,j} - \mathcal{S}(\emptyset)| < 2\epsilon$. The same proof strategy can be applied to $i > N$. Since $\epsilon$ can be arbitrarily small, the results show that all node pairs have the same proximity $\mathcal{S}(\emptyset)$, which leads to a contraction and finishes our proof. □

## A.2 THEOREM 2

Here we formulate and prove Theorem 2. Note that some notations and definitions are introduced in Appendix A.1.

**Theorem 2.** *For the walk-based proximity $\mathbf{S} = \tilde{\mathbf{A}}^K (\tilde{\mathbf{A}}^K)^T$, SMP can preserve the proximity with high probability if the dimensionality of the stochastic matrix is sufficiently large, i.e., $\forall \epsilon > 0, \forall \delta > 0$, there $\exists d_0$ so that any $d > d_0$:*

$$P\left(\left|\mathbf{S}_{i,j} - \mathcal{F}_{de}\left(\mathbf{H}_{i,:}, \mathbf{H}_{j,:}\right)\right| < \epsilon\right) > 1 - \delta, \tag{20}$$

*where $\mathbf{H}$ are the node representation obtained from SMP in Eq. (11). The result holds for any stochastic matrix and thus is regardless of whether $\mathbf{E}$ is fixed or resampled during each epoch.*

*Proof.* Our proof is mostly based on the standard random projection theory. Firstly, since we have proven in Theorem 1 that the permutation-equivariant representations cannot preserve any walk-based proximity, here we prove that we can preserve the proximity only using $\tilde{\mathbf{E}}$, which can be

easily achieved by ignoring $\mathbf{H}^{(L)}$ in $\mathcal{F}_{\text{output}}([\tilde{\mathbf{E}}, \mathbf{H}^{(L)}])$, e.g., if we set $\mathcal{F}_{\text{output}}$ as a linear function, the model can learn to set the corresponding weights for $\mathbf{H}^{(L)}$ as all-zeros.

We set the decoder function as a normalized inner product:

$$\mathcal{F}_{\text{de}}\left(\mathbf{H}_{i,:}, \mathbf{H}_{j,:}\right) = \frac{1}{d}\mathbf{H}_{i,:}\mathbf{H}_{j,:}^T. \tag{21}$$

Then, denoting $\mathbf{a}_i = \tilde{\mathbf{A}}_{i,:}^K$ and recalling $\tilde{\mathbf{E}} = \tilde{\mathbf{A}}^K\mathbf{E}$, we have:

$$|\mathbf{S}_{i,j} - \mathcal{F}_{\text{de}}\left(\mathbf{H}_{i,:}, \mathbf{H}_{j,:}\right)| = |\mathbf{a}_i\mathbf{a}_j^T - \frac{1}{d}\tilde{\mathbf{E}}_{i,:}\tilde{\mathbf{E}}_{j,:}^T| = |\mathbf{a}_i\mathbf{a}_j^T - \mathbf{a}_i\frac{1}{d}\mathbf{E}\mathbf{E}^T\mathbf{a}_j^T|. \tag{22}$$

Since $\mathbf{E}$ is a Gaussian random matrix, from the Johnson-Lindenstrauss lemma (Vempala, 2005) (in the inner product preservation forum, e.g., see Corollary 2.1 and its proof in (Sham & Greg, 2020)), $\forall 0 < \epsilon' < \frac{1}{2}$, we have:

$$P\left(|\mathbf{a}_i\mathbf{a}_j^T - \mathbf{a}_i\frac{1}{d}\mathbf{E}\mathbf{E}^T\mathbf{a}_j^T| \le \frac{\epsilon'}{2}(\|\mathbf{a}_i\| + \|\mathbf{a}_j\|)\right) > 1 - 4e^{-\frac{(\epsilon'^2 - \epsilon'^3)d}{4}}. \tag{23}$$

By setting $\epsilon' = \frac{\epsilon}{\max_i \|\mathbf{a}_i\|}$, we have $\epsilon > \frac{\epsilon'}{2}(\|\mathbf{a}_i\| + \|\mathbf{a}_j\|)$ and:

$$P\left(|\mathbf{S}_{i,j} - \mathcal{F}_{\text{de}}\left(\mathbf{H}_{i,:}, \mathbf{H}_{j,:}\right)| < \epsilon\right) > 1 - 4e^{-\frac{(\frac{\epsilon}{\max_i \|\mathbf{a}_i\|}^2 - \frac{\epsilon}{\max_i \|\mathbf{a}_i\|}^3)d}{4}}, \tag{24}$$

which leads to the theorem by solving and setting $d_0$ as follows:

$$4e^{-\frac{(\frac{\epsilon}{\max_i \|\mathbf{a}_i\|}^2 - \frac{\epsilon}{\max_i \|\mathbf{a}_i\|}^3)d_0}{4}} = \delta \Rightarrow d_0 = \frac{4\log\frac{4}{\delta}\left(\max_i \|\mathbf{a}_i\|\right)^3}{\epsilon^2 \max_i \|\mathbf{a}_i\| - \epsilon^3}. \tag{25}$$

$\square$

## A.3  Theorem 3

Here we formulate and prove Theorem 3. Note that some notations and definitions are introduced in Appendix A.1.

**Theorem 3.** *For any length-$L$ walk-based proximity, i.e.,*

$$\mathbf{S}_{i,j} = \mathcal{S}\left(\{v_i \rightsquigarrow v_j\}\right) = \mathcal{S}\left(\{v_i \rightsquigarrow v_j | len(v_i \rightsquigarrow v_j) \le L\}\right),$$

*where $len(\cdot)$ is the length of a walk, there exists an SMP variant in Eq. (10) with $\mathcal{F}_{GNN}\left(\mathbf{A}, \mathbf{E}; \mathbf{W}\right)$ containing $L$ layers (including the input layer) to preserve that proximity if the following conditions hold: (1) The stochastic matrix $\mathbf{E}$ contains unique signals for different nodes, i.e. $\mathbf{E}_{i,:} \ne \mathbf{E}_{j,:}, \forall i \ne j$. (2) The message-passing and updating functions in learning $\tilde{\mathbf{E}}$ are bijective. (3) The decoder function $\mathcal{F}_{de}(\cdot)$ also takes $\mathbf{E}$ as inputs and is universal approximation.*

*Proof.* Similar as Theorem 2, we only utilize $\tilde{\mathbf{E}}$ during our proof. We use $\mathbf{e}_i^{(l)}, 0 \le l < L$ to denote the node representations in the $l^{th}$ layer of $\mathcal{F}_{\text{GNN}}\left(\mathbf{A}, \mathbf{E}; \mathbf{W}\right)$, i.e., $\mathbf{e}_i^{(0)} = \mathbf{E}_{i,:}$ and $\mathbf{e}_i^{(L-1)} = \tilde{\mathbf{E}}_{i,:}$. Our proof strategy is to show that the stochastic node representations can remember all the information about the walks.

Firstly, as the message-passing and updating function are bijective by assumption, we can recover from the node representations in each layer all their neighborhood representations in the previous layer. Specifically, there exist $\mathcal{F}^{(l)}(\cdot), 1 \le l < L$ such that:

$$\mathcal{F}^{(l)}\left(\mathbf{e}_i^{(l)}\right) = \left[\mathbf{e}_i^{(l-1)}, \left\{\mathbf{e}_j^{(l-1)}, j \in \mathcal{N}_i\right\}\right]^6. \tag{26}$$

For notation conveniences, we split the function into two parts, one for the node itself and the other for its neighbors:

$$\mathcal{F}_{\text{self}}^{(l)}\left(\mathbf{e}_i^{(l)}\right) = \mathbf{e}_i^{(l-1)},$$
$$\mathcal{F}_{\text{neighbor}}^{(l)}\left(\mathbf{e}_i^{(l)}\right) = \left\{\mathbf{e}_j^{(l-1)}, j \in \mathcal{N}_i\right\}. \tag{27}$$

---

[6]To let $\mathcal{F}^{(l)}(\cdot)$ output a set with arbitrary lengths, we can adopt sequence-based models such an LSTM.

For the first function, if we successively apply such functions from the $l^{th}$ layer to the input layer, we can recover the input features of the GNN, i.e., $\mathbf{E}$. Since the stochastic matrix $\mathbf{E}$ contains a unique signal for different nodes, we can decode the node ID from $\mathbf{e}_i^{(0)}$, i.e., there exists $\mathcal{F}_{\text{self}}^{(0)}\left(\mathbf{e}_i^{(0)}; \mathbf{E}\right) = i$. For brevity, we denote applying such $l + 1$ functions to get the node ID as

$$\mathcal{F}_{\text{self}}^{(0:l)}\left(\mathbf{e}_i^{(l)}\right) = \mathcal{F}_{\text{self}}^{(0)}\left(\mathcal{F}_{\text{self}}^{(1)}\left(... \left(\mathcal{F}_{\text{self}}^{(l)}\left(\mathbf{e}_i^{(l)}\right)\right)\right); \mathbf{E}\right) = i. \tag{28}$$

For the second function, we can apply $\mathcal{F}_{\text{neighbor}}^{(l-1)}$ to the decoded vector set so that we can recover their neighborhood representations in the $(l-2)^{th}$ layer, etc.

Next, we show that for $\mathbf{e}_j^{(l-1)}$, there exists a length-$l$ walk $v_i \rightsquigarrow v_j = (v_{a_1}, v_{a_2}, ..., v_{a_l})$, where $v_{a_1} = v_i$, $v_{a_l} = v_j$ if and only if $\mathcal{F}_{\text{self}}^{(0:l-1)}\left(\mathbf{e}_j^{(l-1)}\right) = a_l = j$ and there exists $\mathbf{e}^{(l-2)}, ..., \mathbf{e}^{(0)}$ such that:

$$\begin{aligned}
\mathbf{e}^{(l-2)} &\in \mathcal{F}_{\text{neighbor}}^{(l-1)}\left(\mathbf{e}_j^{(l-1)}\right), \mathcal{F}_{\text{self}}^{(0:l-2)}\left(\mathbf{e}^{(l-2)}\right) = a_{l-1}, \\
\mathbf{e}^{(l-3)} &\in \mathcal{F}_{\text{neighbor}}^{(l-2)}\left(\mathbf{e}^{(l-2)}\right), \mathcal{F}_{\text{self}}^{(0:l-3)}\left(\mathbf{e}^{(l-3)}\right) = a_{l-2}, \\
&\qquad\qquad ... \\
\mathbf{e}^{(0)} &\in \mathcal{F}_{\text{neighbor}}^{(1)}\left(\mathbf{e}^{(1)}\right), \mathcal{F}_{\text{self}}^{(0:0)}\left(\mathbf{e}^{(0)}\right) = a_1 = i.
\end{aligned} \tag{29}$$

This result is easily verified as:

$$\begin{aligned}
(v_{a_1}, v_{a_2}, ..., v_{a_l}) \text{ is a walk} &\Leftrightarrow \mathcal{E}_{a_i, a_j} = \mathcal{E}_{a_j, a_i} = 1 \Leftrightarrow a_i \in \mathcal{N}_{a_{i+1}}, \forall 1 \le i < l \\
&\Leftrightarrow \exists \mathbf{e}^{(i-1)} \in \mathcal{F}_{\text{neighbor}}^{(i)}\left(\mathbf{e}^{(i)}\right), \mathcal{F}_{\text{self}}^{(0:i-1)}\left(\mathbf{e}^{(i-1)}\right) = a_i, \forall 1 \le i < l.
\end{aligned} \tag{30}$$

Note that all the information is encoded in $\tilde{\mathbf{E}}$, i.e., we can decode $\{v_i \rightsquigarrow v_j | \text{len}(v_i \rightsquigarrow v_j) \le L\}$ from $\mathbf{e}_j^{(L-1)}$ by successively applying $\mathcal{F}_{\text{self}}^{(l)}(\cdot), \mathcal{F}_{\text{neighbor}}^{(l)}(\cdot)$. We can also apply $\mathcal{F}_{\text{self}}^{(0:L-1)}$ to $\mathbf{e}_i^{(L-1)}$ to get the start node ID $i$. Putting it together, we have:

$$\mathcal{F}\left(\mathbf{e}_j^{(L-1)}, \mathbf{e}_i^{(L-1)}\right) = \{v_i \rightsquigarrow v_j | \text{len}(v_i \rightsquigarrow v_j) \le L\}, \tag{31}$$

where $\mathcal{F}(\cdot)$ is composed of $\mathcal{F}_{\text{self}}^{(l)}(\cdot), 0 \le l < L$ and $\mathcal{F}_{\text{neighbor}}^{(l)}(\cdot), 1 \le l < L$. Applying the proximity function $\mathcal{S}(\cdot)$, we have:

$$\mathcal{S}\left(\mathcal{F}\left(\mathbf{e}_j^{(L-1)}, \mathbf{e}_i^{(L-1)}\right)\right) = \mathbf{S}_{i,j}. \tag{32}$$

We finish the proof by setting the real decoder function $\mathcal{F}_{\text{de}}(\cdot)$ to arbitrarily approximate this desired function $\mathcal{S}(\mathcal{F}(\cdot, \cdot))$ under the universal approximation assumption. □

# B    ADDITIONAL EXPERIMENTAL RESULTS

## B.1    GRAPH RECONSTRUCTION

To verify that our proposed SMP can indeed preserve node proximities, we conduct experiments of graph reconstruction (Wang et al., 2016), i.e., using the node representations learned by GNNs to reconstruct the edges of the graph. Graph reconstruction corresponds to the first-order proximity between nodes, i.e., whether two nodes directly have a connection, which is the most straightforward node proximity (Tang et al., 2015). Specifically, following Section 5.2, we adopt the inner product classifier $\text{Sigmoid}(\mathbf{H}_i^T \mathbf{H}_j)$ and use AUC as the evaluation metric. To control the impact of node features (i.e., since many graphs exhibit assortative mixing, even models only using node features can reconstruct the edges to a certain extent), we do not use node features for all the models.

We report the results in Table 6. The results show that SMP greatly outperforms permutation-equivariant GNNs such as GCN and GAT in graph reconstruction, clearly demonstrating that SMP can better preserve node proximities. PGNN shows highly competitive results as SMP. However, similar to other tasks, the intensive memory usage makes PGNN unable to handle medium-scale graphs such as Physics and PubMed.

Table 6: The results of graph reconstruction measured in AUC (%). The best and the second-best results for each dataset, respectively, are in bold and underlined. OOM represents out of memory.

| Model | Grid | Communities | Email | CS | Physics | PPI | Cora | CiteSeer | PubMed |
|---|---|---|---|---|---|---|---|---|---|
| SGC | 74.8±0.4 | 65.4±1.6 | 71.6±0.3 | 66.7±0.1 | 66.2±0.0 | 76.3±0.2 | 56.7±9.7 | 58.5±0.1 | 71.9±0.1 |
| GCN | 73.0±0.3 | 63.7±1.2 | 72.5±0.4 | 75.5±0.4 | 76.8±0.4 | 79.2±0.4 | 68.2±3.9 | 69.8±8.0 | 77.2±2.1 |
| GAT | 59.6±1.2 | 52.9±1.1 | 56.9±1.9 | 57.0±1.4 | 59.1±0.7 | 61.1±1.9 | 57.8±1.0 | 63.2±1.5 | 58.8±0.8 |
| PGNN | **99.4±0.1** | 97.7±0.1 | 85.6±0.8 | **97.2±0.6** | OOM | 85.2±0.6 | **98.1±0.6** | **99.7±0.1** | OOM |
| SMP-Identity | 99.2±0.1 | 97.5±0.1 | 80.0±0.3 | 77.1±2.3 | 73.7±0.3 | 79.5±0.2 | 89.7±5.7 | 97.1±0.8 | 77.0±0.1 |
| SMP-Linear | 99.1±0.1 | **97.8±0.1** | **86.7±0.2** | 96.3±0.2 | **95.5±0.2** | **85.5±0.1** | 96.3±0.1 | 98.2±0.1 | **95.8±0.2** |

## B.2 PAIRWISE NODE CLASSIFICATION

Besides standard node classification experiments reported in Section 5.3, we follow (You et al., 2019) and experiment on pairwise node classification, i.e., predicting whether two nodes have the same label. Compared with standard node classification, pairwise node classification focuses more on the relations between nodes and thus requires the model to be proximity-aware to perform well.

Similar to link prediction, we split the positive samples (i.e., node pairs with the same label) into an 80%-10%-10% training-validation-testing set with an equal number of randomly sampled negative pairs. For large graphs, since the possible positive samples are intractable (i.e. $O(N^2)$), we use a random subset. Since we also need node labels as the ground-truth, we only conduct pairwise node classification on datasets when node labels are available. We also exclude the results of PPI since the dataset is multi-label and cannot be used in a pairwise setting (You et al., 2019). Similar to Section 5.2, we adopt a simple inner product classifier and use AUC as the evaluation metric.

The results are shown in Table 7. We observe consistent results as link prediction in Section 5.2, i.e., SMP reports the best results on four datasets and the second-best results on the other three datasets. These results again verify that SMP can effectively preserve and utilize node proximities when needed while retaining comparable performance when the tasks are more permutation-equivariant like, e.g., on CS and Physics.

Table 7: The results of pairwise node classification tasks measured in AUC (%). The best results and the second-best results for each dataset, respectively, are in bold and underlined.

| Model | Communities | Email | CS | Physics | Cora | CiteSeer | PubMed |
|---|---|---|---|---|---|---|---|
| SGC | 67.4±2.4 | 56.3±5.4 | **99.8±0.0** | 99.6±0.0 | 99.2±0.3 | **95.5±0.7** | 92.3±0.3 |
| GCN | 64.9±2.3 | 55.0±5.7 | 96.8±0.7 | **99.7±0.1** | 97.7±0.6 | 92.9±1.2 | **94.8±0.4** |
| GAT | 52.5±1.3 | 47.7±2.7 | 95.2±0.6 | 96.3±0.2 | 91.6±0.7 | 73.6±2.7 | 87.1±0.2 |
| PGNN | 98.6±0.5 | 63.3±5.5 | 90.0±0.5 | Out of memory | 85.5±1.2 | 49.8±1.8 | Out of memory |
| SMP-Identity | **98.8±0.5** | 56.9±4.1 | 99.7±0.0 | 99.6±0.0 | 99.2±0.2 | 95.2±1.1 | 91.9±0.3 |
| SMP-Linear | **98.8±0.5** | **74.5±4.1** | **99.8±0.0** | 99.6±0.0 | **99.3±0.3** | 95.3±0.4 | 93.4±0.2 |

## B.3 COMPARISON WITH USING IDS

We further compare SMP with augmenting GNNs using a one-hot encoding of node IDs, i.e., the identity matrix. Intuitively, since the IDs of nodes are unique, such a method does not suffer from the automorphism problem and should also enable GNNs to preserve node proximities. However, theoretically speaking, using such a one-hot encoding has two major problems. Firstly, the dimensionality of the identity matrix is $N \times N$, and thus the number of parameters in the first message-passing layer is also on the order of $O(N)$. Therefore, the method will inevitably be computationally expensive and may not be scalable to large-scale graphs. A large number of parameters will also more likely lead to the overfitting problem. Secondly, the node IDs are not transferable across different graphs, i.e., the node $v_1$ in one graph and the node $v_1$ in another graph do not necessarily share a similar meaning. But as the parameters in the message-passings depend on the node IDs (since they are input features), such a mechanism cannot handle inductive tasks well.[7]

---

[7]One may question whether SMP is transferable across different graphs since the stochastic features are independently drawn. Empirically, we find that SMP reports reasonably well results on inductive datasets such

Table 8: The results of comparing SMP with using one-hot IDs in GCNs. OOM represents out of memory. — represents the task is unavailable.

| Task | Model | Grid | Communities | Email | CS | Physics | PPI | Cora | CiteSeer | PubMed |
|------|-------|------|-------------|-------|-----|---------|-----|------|----------|--------|
| Link Prediction | $GCN_{onehot}$ | 91.5±2.1 | 98.3±0.7 | 71.2±3.5 | 93.1±1.3 | OOM | 78.6±0.3 | 86.8±1.5 | 81.7±1.1 | 89.4±0.5 |
| | SMP-Linear | 73.6±6.2 | 97.7±0.5 | 75.7±5.0 | 96.7±0.1 | 96.1±0.1 | 81.9±0.3 | 92.7±0.7 | 92.6±1.0 | 95.4±0.2 |
| Pairwise Node Classification | $GCN_{onehot}$ | — | 98.9±0.5 | 67.3±5.6 | 97.6±0.2 | OOM | — | 98.2±0.3 | 94.4±1.2 | 98.9±0.1 |
| | SMP-Linear | — | 98.8±0.5 | 74.5±4.1 | 99.8±0.0 | 99.6±0.0 | — | 99.3±0.3 | 95.3±0.4 | 93.4±0.2 |
| Node Classification | $GCN_{onehot}$ | — | 99.6±1.0 | — | 86.9±1.5 | OOM | — | 77.6±1.1 | 57.7±5.8 | 74.9±0.6 |
| | SMP-Linear | — | 99.9±0.3 | — | 91.5±0.8 | 93.1±0.8 | — | 80.9±0.8 | 68.2±1.0 | 76.5±0.8 |

We also empirically compare such a method with SMP and report the results in Table 8. The results show that SMP-Linear outperforms $GCN_{onehot}$ in most cases. Besides, $GCN_{onehot}$ fails to handle Physics, which is only a medium-scale graph, due to the heavy memory usage. One surprising result is that $GCN_{onehot}$ outperforms SMP-Linear on Grid, the simulated graph where nodes are placed on a $20 \times 20$ grid. A plausible reason is that since the edges in Grid follow a specific rule, using a one-hot encoding gives $GCN_{onehot}$ enough flexibility to learn and remember the rules, and the model does not overfit because the graph has a rather small scale.

## B.4 ADDITIONAL LINK PREDICTION RESULTS

We further report the results of link prediction on three GNN benchmarks: Cora, CiteSeer, and PubMed. The results are shown in Table 9. The results show similar trends as other datasets presented in Section 5.2, i.e., SMP reports comparable results as other permutation-equivariant GNNs while PGNN fails to handle the task well.

Table 9: The results of the link prediction task measured in AUC (%). The best results and the second-best results for each dataset, respectively, are in bold and underlined.

| Model | Cora | CiteSeer | PubMed |
|-------|------|----------|--------|
| SGC | **93.6±0.6** | **94.7±0.8** | **95.8±0.2** |
| GCN | 90.6±1.0 | 78.2±1.7 | 92.4±0.9 |
| GAT | 88.5±1.2 | 87.8±1.0 | 89.2±0.8 |
| PGNN | 75.4±2.3 | 70.6±1.1 | Out of memory |
| SMP-Identity | 93.0±0.6 | 92.9±0.5 | 94.5±0.3 |
| SMP-Linear | 92.7±0.7 | 92.6±1.0 | 95.4±0.2 |
| SMP-MLP | 82.8±0.9 | 80.7±1.1 | 88.0±0.6 |
| SMP-Linear-$GCN_{feat}$ | 86.7±1.4 | 81.1±1.4 | 90.5±0.6 |
| SMP-Linear-$GCN_{both}$ | 80.1±2.5 | 80.0±2.0 | 81.1±2.0 |

## B.5 ADDITIONAL ABLATION STUDIES

We report the ablation study results for the node classification task and pairwise node classification task in Table 10 and Table 11, respectively. The results again show that SMP-Linear generally achieves good-enough results on the majority of the datasets and adding non-linearities does not necessarily lift the performance of SMP.

We also compare whether the stochastic signals $\mathbf{E}$ are fixed or not during different training epochs for our proposed SMP. For brevity, we only report the results for the link prediction task in Table 12. The results show that fixing $\mathbf{E}$ usually leads to better results on transductive datasets (recall that datasets except Email and PPI are transductive) and resampling $\mathbf{E}$ leads to better results on inductive datasets in general. The results are consistent with our analysis in Section 4.1.

---

as Email and PPI. One plausible reason is that since the proximities of nodes are preserved even the random features per se are different (see Theorem 2), all subsequent parameters based on proximities can be transferred.

Table 10: The ablation study of different SMP variants for the node classification task. The best results and the second-best results are in bold and underlined, respectively.

| Model | Communities | CS | Physics | Cora | CiteSeer | PubMed |
|---|---|---|---|---|---|---|
| SMP-Linear | 99.9±0.3 | **91.5±0.8** | **93.1±0.8** | **80.9±0.8** | **68.2±1.0** | 76.5±0.8 |
| SMP-MLP | **100.0±0.2** | 90.1±0.5 | 92.3±0.8 | 79.3±0.8 | 67.0±1.5 | 76.8±0.9 |
| SMP-Linear-GCN$_{\text{feat}}$ | **100.0±0.0** | 89.8±0.7 | 92.9±0.8 | 78.9±1.2 | 67.8±0.6 | **77.3±0.6** |
| SMP-Linear-GCN$_{\text{both}}$ | **100.0±0.2** | 77.4±4.2 | 87.1±3.5 | 69.2±2.5 | 49.8±3.1 | 68.1±4.1 |

Table 11: The ablation study of different SMP variants for the pairwise node classification task. The best results and the second-best results are in bold and underlined, respectively.

| Model | Communities | Email | CS | Physics | Cora | CiteSeer | PubMed |
|---|---|---|---|---|---|---|---|
| SMP-Identity | 98.8±0.5 | 56.9±4.1 | 99.7±0.0 | **99.6±0.0** | 99.2±0.2 | 95.2±1.1 | 91.9±0.3 |
| SMP-Linear | 98.8±0.5 | **74.5±4.1** | **99.8±0.0** | **99.6±0.0** | **99.3±0.3** | **95.3±0.4** | 93.4±0.2 |
| SMP-MLP | 98.7±0.3 | 65.4±6.3 | 94.3±0.6 | 97.6±0.4 | 90.3±3.0 | 67.7±13.7 | 93.4±0.4 |
| SMP-Linear-GCN$_{\text{feat}}$ | **99.0±0.4** | 60.2±9.3 | 95.6±0.7 | 98.3±0.7 | 96.1±0.7 | 88.8±1.6 | **94.8±0.2** |
| SMP-Linear-GCN$_{\text{both}}$ | 98.8±0.4 | 61.6±6.0 | 95.2±0.7 | 97.8±0.8 | 94.3±1.9 | 83.5±3.9 | 94.1±0.7 |

### B.6 COMPARISON OF PERMUTATION-EQUIVARIANT GNNs FOR LINK PREDICTION

To investigate the performance of linear and non-linear variants of permutation-equivariant GNNs for the link prediction task, we additionally report both the training accuracies and the testing accuracies of SGC, GCN, and GAT in Table 13. Notice that to ensure a fair comparison, we do not adopt the early stopping strategy here so that different models have the same number of training epochs (otherwise, if a model tends to overfit, the early stopping strategy will terminate the training process when the number of training epochs is small and result in a spurious underfitting phenomena).

The results show that non-linear variants of GNNs (GCN and GAT) are more likely to overfit, i.e., the margins between the training accuracies and the testing accuracies are usually larger, than the linear variant SGC. Besides, though possessing extra model expressiveness, non-linear GNNs are also difficult to train, i.e., the training accuracies of GCN and GAT are not necessarily higher than SGC. The results are consistent with the literature Wu et al. (2019); He et al. (2020).

## C EXPERIMENTAL DETAILS FOR REPRODUCIBILITY

### C.1 DATASETS

- **Grid** (You et al., 2019): A simulated 2D grid graph with size $20 \times 20$ and no node feature.

- **Communities** (You et al., 2019): A simulated caveman graph (Watts, 1999) composed of 20 communities with each community containing 20 nodes. The graph is perturbed by rewiring 1% edges randomly. It has no node feature and the label of each node indicates which community the node belongs to.

- **Email**[8] (You et al., 2019): Seven real-world email communication graphs. Each graph has six communities and each node has an integer label indicating the community the node belongs to.

- **Coauthor Networks**[9] (Shchur et al., 2018): Two networks from Microsoft academic graph in *CS* and *Physics* with their nodes representing authors and edges representing co-authorships between authors. The node features are embeddings of the paper keywords of the authors.

- **PPI**[8] (Hamilton et al., 2017): 24 protein-protein interaction networks. Each node has a 50-dimensional feature vector.

- **PPA**[10] (Hu et al., 2020): A network representing biological associations between proteins from 58 different species. The node features are one-hot vectors of the species that the proteins are taken from.

---

[8] https://github.com/JiaxuanYou/P-GNN/tree/master/data
[9] https://github.com/shchur/gnn-benchmark/tree/master/data/npz/
[10] https://snap.stanford.edu/ogb/data/linkproppred/ppassoc.zip

Table 12: The results of comparing whether the stochastic signals $\mathbf{E}$ are fixed or not during different training epochs for the link prediction task. The better of the two results are in bold.

| Model | $\mathbf{E}$ | Grid | Communities | CS | Physics | Email | PPI |
|---|---|---|---|---|---|---|---|
| SMP-Identity | Fixed | 55.1±4.8 | **98.0±0.7** | **96.5±0.1** | 96.5±0.1 | **75.9±3.9** | 80.4±0.4 |
| | Not Fixed | **55.2±4.1** | 97.6±0.7 | 96.4±0.1 | 96.5±0.1 | 72.9±5.1 | **81.0±0.2** |
| SMP-Linear | Fixed | **73.6±6.2** | **97.7±0.5** | **96.7±0.1** | 96.1±0.1 | 71.3±3.9 | 71.5±0.7 |
| | Not Fixed | 64.4±2.9 | 97.4±0.1 | 96.2±0.1 | 96.1±0.1 | **75.7±5.0** | **81.9±0.3** |

Table 13: The results of SGC, GCN, and GAT for the link prediction task. Both the training accuracies and the testing accuracies are reported.

| Method | Results | Grid | Communities | Email | CS | Physics | PPI | Cora | CiteSeer | PubMed |
|---|---|---|---|---|---|---|---|---|---|---|
| SGC | Train | 52.4±0.5 | 50.4±0.5 | 68.4±1.0 | 98.3±0.0 | 97.7±0.0 | 84.9±0.4 | 99.5±0.0 | 99.9±0.0 | 98.9±0.0 |
| | Test | 51.6±2.9 | 49.2±1.6 | 67.0±9.3 | 96.5±0.1 | 96.5±0.1 | 78.8±0.7 | 92.5±0.7 | 94.0±0.5 | 95.6±0.3 |
| GCN | Train | 51.3±1.0 | 50.0±0.6 | 68.9±4.0 | 95.3±0.3 | 95.1±0.3 | 76.1±0.6 | 96.5±1.0 | 77.3±1.3 | 93.9±1.6 |
| | Test | 54.6±4.3 | 49.2±1.1 | 66.0±3.1 | 93.2±0.3 | 93.8±0.3 | 74.9±0.6 | 89.2±0.3 | 76.1±2.5 | 90.6±1.2 |
| GAT | Train | 47.5±1.4 | 49.6±0.3 | 52.2±3.6 | 95.7±0.9 | 96.4±0.5 | 82.3±0.3 | 97.2±0.5 | 99.3±0.1 | 98.3±0.1 |
| | Test | 50.8±6.0 | 50.8±2.1 | 47.6±4.4 | 91.0±1.1 | 93.9±0.3 | 78.5±0.4 | 75.7±1.7 | 81.0±0.7 | 83.7±1.3 |

- **Cora**, **CiteSeer**, **PubMed**[11] (Yang et al., 2016): Three citation graphs where nodes correspond to papers and edges correspond to citations between papers. The node features are bag-of-words and the node labels are the ground truth topics of the papers.

We summarize the statistics of datasets in Table 14.

## C.2 HYPER-PARAMETERS

We use the following hyper-parameters:

- All datasets except PPA: we uniformly set the number of layers for all the methods as 2, i.e., 2 message-passing steps, and set the dimensionality of hidden layers as 32, i.e., $\mathbf{H}^{(l)} \in \mathbb{R}^{N \times 32}$, for all $1 \leq l \leq L$ (for GAT, we use 4 heads with each head containing 8 units). We use Adam optimizer with an initial learning rate of 0.01 and decay the learning rate by 0.1 at epoch 200. The weight decay is 5e-4. We train the model for 1,000 epochs and evaluate the model every 5 epochs. We adopt an early-stopping strategy by reporting the testing performance at the epoch which achieves the best validation performance. For SMP, the dimensionality of the stochastic matrix is $d = 32$. For P-GNN, we use the P-GNN-F version, which uses the truncated 2-hop shortest path distance instead of the exact shortest distance.

- PPA: as suggested in the original paper (Hu et al., 2020), we set the number of GNN layers as 3 with each layer containing 256 hidden units and add a three-layer MLP after taking the Hadamard product between pair-wise node embeddings as the predictor, i.e., $\mathrm{MLP}(\mathbf{H}_i \odot \mathbf{H}_j)$. We use Adam optimizer with an initial learning rate of 0.01. We set the number of epochs for training as 40, evaluate the results on validation sets every epoch, and report the testing results using the model with the best validation performance. We also found that the dataset had issues with exploding gradients and adopt a gradient clipping strategy by limiting the maximum p2-norm of gradients as 1.0. The dimensionality of the stochastic matrix in SMP is $d = 64$.

## C.3 HARDWARE AND SOFTWARE CONFIGURATIONS

All experiments are conducted on a server with the following configurations.

- Operating System: Ubuntu 18.04.1 LTS
- CPU: Intel(R) Xeon(R) CPU E5-2699 v4 @ 2.20GHz
- GPU: NVIDIA TESLA M40 with 12 GB of memory

---

[11]https://github.com/kimiyoung/planetoid/tree/master/data

Table 14: The statistics of the datasets. For datasets with more than one graph, #Nodes and #Edges are summed over all the graphs and the experiments are conducted in an inductive setting.

| Dataset | #Graphs | #Nodes | #Edges | #Features | #Classes |
|---|---|---|---|---|---|
| Grid | 1 | 400 | 760 | - | - |
| Communities | 1 | 400 | 3,800 | - | 20 |
| Email | 7 | 1,005 | 25,571 | - | 42 |
| CS | 1 | 18,333 | 81,894 | 6,805 | 15 |
| Physics | 1 | 34,493 | 247,962 | 8,415 | 5 |
| PPI | 24 | 56,944 | 818,716 | 50 | - |
| PPA | 1 | 576,289 | 30,326,273 | 58 | - |
| Cora | 1 | 2,708 | 5,429 | 1,433 | 7 |
| CiteSeer | 1 | 3,327 | 4,732 | 3,703 | 6 |
| PubMed | 1 | 19,717 | 44,338 | 500 | 3 |

- Software: Python 3.6.8, PyTorch 1.4.0, PyTorch Geometric 1.4.3, NumPy 1.18.1, Cuda 10.1

