# OpenReview forum: "A Simple and General Graph Neural Network with Stochastic Message Passing"
_ICLR.cc/2021/Conference — Reject_

### Official Review · AnonReviewer3 · 2020-10-16
**Methods are known, results are trivial.**

**Rating:** 3
**Confidence:** 5

**Review:**

While the paper is easy to follow, I found all the results in this paper trivial and already-known.

Pros:
1. The paper is well-written and easy to follow.

Cons:
1. The entire Section 3 is not novel. It is a mixture of the preliminaries of GNNs and the trivial results of Corollary 1 and Theorem 1. Moreover, Theorem 1 is not rigorous. What if walk-based proximity is just a constant function? What if a given graph does not contain any automorphism?
2. “Preserve walk-based similarity” is not rigorously defined. It seems that it just means all nodes have different embeddings.
3. The proposed model is not permutation-equivariant after adding Gaussian noise. It is trivial that the model becomes permutation-equivariant when the Gaussian noise is ignored (because the model just reduces to an ordinary GNN).
4. The claim that SMP preserves walk-based proximity is trivial and just relies on the fact that randomly-sampled vectors from a Gaussian distribution are different from each other.
5. The idea of using node identifiers (essentially equivalent to the Gaussian noise) to make GNNs position-aware is not new. In fact, this idea is clearly mentioned in the P-GNN paper already (Section 6.2 of [1] “for inductive tasks, augmenting node attributes with one-hot identifiers restricts a model’s generalization ability”).
6. In Section 5.3, it is unclear why the OGB node classification datasets are not used.

[1] https://arxiv.org/abs/1906.04817

---

> ### Author Response · Authors · 2020-11-18
> **To Reviewer 3**
>
> Thank you for your detailed comments. Here are our responses to your questions.
>
> Q1-1. The entire Section 3 is not novel. It is a mixture of the preliminaries of GNNs and the trivial results of Corollary 1 and Theorem 1.
> A1-1. Section 3 provides the necessary preliminaries of GNNs and several key definitions, which is important to keep our paper self-contained. Besides, we give novel proof that the existing permutation-equivariant GNNs cannot preserve node proximities, which lays the foundations for further developing our proposed method.
> Q1-2. Moreover, Theorem 1 is not rigorous. What if walk-based proximity is just a constant function?
> A1-2. Thanks for the suggestion. We have clarified that we only consider non-trivial walk-based proximity in the revised version. Please refer to Theorem 1 of the revised manuscript for details.
> Q1-3. What if a given graph does not contain any automorphism?
> A1-3: As we show in Theorem 1, (non-trivial) automorphism presents an important limitation to the existing GNNs, i.e., preventing GNNs from being proximity-aware. Considering the importance of this limitation, we think it is critical to study this problem even not all graphs contain automorphism.
>
> Q2. “Preserve walk-based similarity” is not rigorously defined. It seems that it just means all nodes have different embeddings.
> A2. We kindly do not agree that preserving walk-based proximity is equivalent to having different node embeddings. The definition was formally provided in Definition 4 in Appendix A.1 due to the page limit and we have moved it into the main paper (Section 3) in the revised version. In a nutshell, a GNN is said to be able to preserve walk-based similarities if we can recover the similarity between two nodes from the corresponding two node embeddings. Thus, the embedding vectors need to encode sufficient information of the walk-based proximity rather than simply being unique node identifiers. Please refer to the revised manuscript for details.
>
> Q3. The proposed model is not permutation-equivariant after adding Gaussian noise. It is trivial that the model becomes permutation-equivariant when the Gaussian noise is ignored (because the model just reduces to an ordinary GNN).
> A3. We kindly do not agree that these theorems should be considered trivial. Our Remark 1 and Corollary 2 indeed show that we can easily recover a permutation-equivariant GNN by ignoring the stochastics representations. Though not using sophisticated proof strategies, these theorems are indispensable to show that our proposed method is able to handle both permutation-equivariant and proximity-aware tasks. On the contrary, though P-GNN can preserve node proximities to a certain extent, it cannot reduce to a permutation-equivariant GNN and thus fail to handle tasks where permutation-equivariance is helpful. We also empirically validate the importance of this result in experiments, i.e., in the node classification task in Section 5.3, our proposed SMP achieves comparable results as GCNs but P-GNN performs poorly. As a result, we believe these theorems are important for our proposed method.
>
> Q4. The claim that SMP preserves walk-based proximity is trivial and just relies on the fact that randomly-sampled vectors from a Gaussian distribution are different from each other.
> A4. We kindly do not agree that the theorem is trivial. If only i.i.d. Gaussian random vectors are adopted as the reviewer suggests, it is impossible to preserve walk-based proximities since these random vectors are independent of the proximity. In fact, we need to properly perform message-passing on those random vectors and carefully decode the node representations to preserve the proximity. We have given and proved the exact procedure for both the linear case (see Theorem 2) and the non-linear case (see Theorem 3).
>
> Q5. The idea of using node identifiers (essentially equivalent to the Gaussian noise) to make GNNs position-aware is not new. In fact, this idea is clearly mentioned in the P-GNN paper already (Section 6.2 of [1] “for inductive tasks, augmenting node attributes with one-hot identifiers restricts a model’s generalization ability”).
> [1] https://arxiv.org/abs/1906.04817
> A5. We humbly do not agree that using Gaussian random vectors is equivalent to using one-hot identifiers. We have compared our proposed method using random vectors with using one-hot IDs as node identifiers in Table 8 in the appendix. The results show that our proposed method generally achieves better results. Besides, using one-hot IDs will drastically increase the number of parameters since the input features of GNNs have a dimensionality of O(n). Besides, we’d like to point out that we have compared with P-GNN as a baseline in all the experiments.

---

> > ### Author Response · Authors · 2020-11-18
> > **To Reviewer 3 (continued)**
> >
> > Q6. In Section 5.3, it is unclear why the OGB node classification datasets are not used.
> > A6: We do not adopt more OGB datasets because we have already adopted ten datasets commonly used in GNN. These datasets already cover a wide spectrum of domains, sizes, and with or without node features, which we believe is more than sufficient to demonstrate our proposed method.

---

> > > ### Comment · AnonReviewer3 · 2020-11-18
> > > **Thank you for your response. Need clarification.**
> > >
> > > I thank the authors for the response. I need some clarification on how SMP works exactly. Does SMP (1) randomly sample the input node embeddings from Gaussian in every forward pass (hence, the computed node embeddings are different in every forward pass), or (2) sample the Gaussian vectors once at the beginning and subsequently use the same ones throughout the training and inference (hence, the computed node embeddings are the same for different forward passes)?
> > >
> > > If (1) is the case, Theorem 3 should not hold since the computed node embeddings are stochastic, while Definition 4 is about the static property (no statement about "with high probability"). I think Definition 4 needs to be modified to take the stochastic nature of the SMP into account.
> > >
> > > If (2) is the case, then all the results are trivial to me, since having fixed unique embeddings for each node + having the very expressive decoder should be able to approximate any functions over a pair of nodes (e.g., shortest path distance). Thus, Definition 4 is trivially attained.
> > >
> > > Re: "Besides, using one-hot IDs will drastically increase the number of parameters since the input features of GNNs have a dimensionality of O(n)."
> > >
> > > I disagree with this. If one-hot IDs are used and the embedding layer is not trained (so the parameters stay the randomly initialized vectors throughout the training), then this is equivalent to SMP of case (2).

---

> > > > ### Author Response · Authors · 2020-11-18
> > > > **Responses**
> > > >
> > > > We thank the reviewer for additional feedbacks. Here are our responses.
> > > >
> > > > Q: How does SMP work?
> > > > A: As we have explained in Section 4.1 (lines 174-181 in the revised version), we fix the Gaussian vector (i.e., (2) as you suggested) for transductive datasets and resample the vector (i.e., (1) as you suggested) for inductive datasets. An ablation study is also reported (Table 12 in appendix B.5) to verify this design.
> > > >
> > > > Q: Why Theorem 3 holds even if the node embeddings are stochastic?
> > > > A: There are 3 conditions for Theorem 3 to hold (shown in lines 575-579 in the revised version): the stochastic matrix contains unique signals for different nodes; the functions in GNNs are bijective; the decoder is very expressive and also takes the stochastic matrix as inputs. Under these conditions, since we only use the stochastic matrix as node identifiers, we can handle stochastic node embeddings (i.e., input different stochastic node embeddings into the the same decoder) without modifying the definition or the theorem.
> > > >
> > > > Q: Is having unique embedding all you need to preserve proximities?
> > > > A: We kindly do not agree that having unique embeddings can approximate any function. Note that in the decoding phase, we cannot use the actual proximity or the graph structure (otherwise, there is no point preserving the proximity, since we already know the proximity or we can recalculate the proximity). More theoretically, [1] shows that having unique node identifiers being one necessary (but not sufficient) condition for GNNs to be universal approximators (specifically, [1] shows that the GNN also needs to be sufficiently expressive). Thus, the theorems cannot be trivially attained. We will further clarify our expressions.
> > > > [1] What graph neural networks cannot learn: depth vs width, ICLR 2020
> > > >
> > > > Q: Regarding one-hot IDs
> > > > A: We agree that using one-hot IDs while fixing the parameters in the first layer as random Gaussians is equivalent to the stochastic part of our proposed method (essentially, this is to replace E in our proposed method with I*E). However, we do not find that such a method is used in any previous GNN, i.e., when referring to using one-hot IDs as inputs, they all assume the first layer is trainable. Besides, our proposed SMP also has a permutation-equivariant GNN and an output function. Thus, we feel that the reviewer's comments provide an alternative understanding of SMP, but do not affect the novelty of our proposed method.

---

> > > > > ### Comment · AnonReviewer3 · 2020-11-18
> > > > > **Thanks for clarification. Further questions.**
> > > > >
> > > > > I thank the authors for clarification, especially on how SMP works. I still have further questions regarding the other answers.
> > > > >
> > > > > Re: The decoder is very expressive and also takes the stochastic matrix as inputs.
> > > > >
> > > > > In my opinion, this is an extremely unrealistic assumption given that Gaussian distribution has support everywhere in real vector space. In practice, I do not think we can learn the decoder such that it preserves node proximity for uncountably-infinitely-many sets of Gaussian random vectors. Theorem 3 definitely needs to be modified to avoid such an assumption.
> > > > >
> > > > > Re: Note that in the decoding phase, we cannot use the actual proximity or the graph structure (otherwise, there is no point preserving the proximity, since we already know the proximity or we can recalculate the proximity)
> > > > >
> > > > > This does not accord with Definition 4, where the authors claim that for each graph, there exists a decoder to approximate the node proximity. This means that the decoder can just be graph-specific; thus, all we need are a unique embedding for each node, and the rest will be taken care of by the graph-specific expressive decoder.
> > > > >
> > > > > Re: However, we do not find that such a method is used in any previous GNN, i.e., when referring to using one-hot IDs as inputs, they all assume the first layer is trainable.
> > > > >
> > > > > To me, making the embedding non-trainable is quite trivial, and I cannot buy the argument. If the authors claim the main methodological novelty is not training the embedding layer, then the authors need to do an ablation study for that. However, I still think the method is already known and trivial.

---

> > > > > > ### Author Response · Authors · 2020-11-19
> > > > > > **Further Responses**
> > > > > >
> > > > > > We thank the reviewer for further responses. Before addressing your detailed comments, we would like to emphasize again that our paper is the first study to maintain both proximity-awareness and permutation-equivariance, which are critical properties of GNNs. Besides the results already shown in the paper, we recently find that such a topic may be very helpful in drug repurposing. For example, the work of [1] by Barabasi et al. shows that GNNs and proximity algorithms lead to complementary discoveries of drug candidates for COVID-19 (see A1-A4, P1-P3 of figure 1 and 2(c) of [1]). Our work provides deep understandings for such a phenomenon since they exactly correspond to proximity-awareness and permutation-equivariance. This is only another example of the potential impact of our paper. Thus, we strongly recommend the reviewer to reconsider whether the results in our paper should all be considered trivial and deserve a score of 1.
> > > > > > [1] Network Medicine Framework for Identifying Drug Repurposing Opportunities for COVID-19, arXiv 2004.07229
> > > > > >
> > > > > > Q: In my opinion, this is an extremely unrealistic assumption given that Gaussian distribution has support everywhere in real vector space. In practice, I do not think we can learn the decoder such that it preserves node proximity for uncountably-infinitely-many sets of Gaussian random vectors. Theorem 3 definitely needs to be modified to avoid such an assumption.
> > > > > > A: (1) Our proof for Theorem 3 is given in Appendix A.2. In the proof, assuming the message-passing functions in GNNs are bijective, embedding vectors with different stochastic Gaussian matrices essentially only affect the node ID decoding step (i.e., line 591). Thus, the same decoder should be able to handle different stochastic representations. Please let us know if you find the proof has any specific issue.
> > > > > > (2) We agree that though Theorem 3 provides important theoretical results, some of its assumptions may be too strong in practice. As explained multiple times in the paper (lines 59-60, 229-232, and the entire Section 5.4), we empirically find that the linear variant of SMP is expressive enough and may be of more practical usage (recall that it has a much simpler decoder, i.e., inner product, and also with theoretical results, i.e., Theorem 2).
> > > > > > (3) We agree that uncountability may bring theoretical issues (similar to concerns raised in [2]). Considering that we can only round numbers to machine precision in practice, we assume the Gaussian vectors are rounded to a countable subset of real numbers in Theorem 3. We have added this assumption (lines 226-227) in the revised paper.
> > > > > > [2] How Powerful Are Graph Neural Networks, ICLR 2019
> > > > > >
> > > > > > Q: This does not accord with Definition 4, where the authors claim that for each graph, there exists a decoder to approximate the node proximity. This means that the decoder can just be graph-specific; thus, all we need are a unique embedding for each node, and the rest will be taken care of by the graph-specific expressive decoder.
> > > > > > A: Thank you for providing this corner case. To make the theorem more rigorous and avoid further misunderstanding, we have slightly revised Definition 4 so that the decoder is explicitly stated as not graph-specific to prevent the trivial cases you mention (we have checked that the rest of our results are not affected by this slight modification). Please refer to the revised manuscript for details (Definition 4 and line 148).
> > > > > >
> > > > > > Q: The authors need to do an ablation study to compare with one-hot IDs.
> > > > > > A: As explained in the paper and our first response, we have reported such a comparison (using stochastic matrix vs. using one-hot IDs) in Table 8 in the appendix. The results show that our proposed method generally achieves better results.
> > > > > >
> > > > > > Q: I still think the method is already known and trivial.
> > > > > > A: We would be appreciated if the reviewer can list the exact reference if there exist other methods/analyses similar to our paper, or our studied problem (maintaining both proximity-awareness and permutation-equivariance in GNNs).

---

> > > > > > > ### Comment · AnonReviewer3 · 2020-11-19
> > > > > > > **Thank you for the response. Not convinced.**
> > > > > > >
> > > > > > > I thank the authors for their response. Unfortunately, after a series of correspondence, I am now confident about my judgment that this paper should not be accepted.
> > > > > > >
> > > > > > > - Q1 is essentially not resolved.
> > > > > > >
> > > > > > > - The change of the claim of Definition 4 (claiming the existence of the universal graph decoder that takes the stochastic node embeddings as input) is rather major and strongly hinges on the unrealistic assumption of my Q1. Furthermore, for transductive scenarios, we are only given a single graph. Therefore, whether we have the universal decoder for any graphs does not really matter. All we need are a unique set of node embeddings + the graph-specific expressive decoder.
> > > > > > >
> > > > > > > - All the essential method (one-hot ID) and insight (limitation of GNNs to capture position information) are already in the P-GNN paper. The newly-derived results are all trivial to me.

---

> > > > > > > > ### Author Response · Authors · 2020-11-20
> > > > > > > > **All questions are addressed**
> > > > > > > >
> > > > > > > > We thank the reviewer for the comments.
> > > > > > > >
> > > > > > > > Q: Regarding Q1 and Unique Node Identifiers
> > > > > > > > A: We do not see what is the specific question of the reviewer since the previous Q1 was addressed by our slight revision of Definition 4 and we also explicitly stated that we do not use the graph structure in the decoder. We also gave the reference [1] showing that it is formally proved that unique node identifiers are not sufficient (but a necessary) condition for GNNs to be universal.  Thus, we consider this question already addressed.
> > > > > > > > [1] What graph neural networks cannot learn: depth vs width, ICLR 2020
> > > > > > > >
> > > > > > > > Q: Differences with P-GNN
> > > > > > > > A: As explained in previous responses, we have compared with P-GNN both methodologically (lines 84-87) and empirically (we have adopted P-GNN as a baseline in all our experiments). Other reviewers also agree that our proposed method is "a significant improvement over P-GNN". So we strongly disagree with your comment that our results should be considered trivial compared to P-GNN.
> > > > > > > >
> > > > > > > > If the reviewer does not have more specific questions, we will consider all your questions have been addressed. The rest we can do is agreeing to differ.

---

> > > > > > > > > ### Comment · AnonReviewer3 · 2020-11-20
> > > > > > > > > **Opinions**
> > > > > > > > >
> > > > > > > > > I thank the authors for addressing my questions. I summarized my opinion below so that our AC can make a final decision.
> > > > > > > > >
> > > > > > > > > Pro:
> > > > > > > > > - Method is simple. I definitely favor a simple method over complex ones, and I agree with R2 (https://openreview.net/forum?id=fhcMwjavKEZ&noteId=C1nSRs79gKk).
> > > > > > > > > - The authors perform a serious effort in evaluating the idea. In practice, the method is working well on the link prediction datasets, but not on the node classification datasets.
> > > > > > > > >
> > > > > > > > > Con:
> > > > > > > > > - From the theory side, I did not learn any new things. All the claims are either already made or (in my opinion) minor extensions from the P-GNN work.
> > > > > > > > > - Although the method is simple, the method is the baseline of P-GNN (Section 6.2 of the paper). I am not sure if we are making progress by proposing the baseline method again, although I also have the intuition that the baseline method is likely to work better than the P-GNN architecture.
> > > > > > > > > - Theoretical claims hinge on unrealistic assumptions on the power of GNNs and the decoder, especially when the node embeddings are stochastic (my Q1).
> > > > > > > > > - Although the authors do use the graph structure in their proof of Theorem 3, in a transductive setting, it still seems that all we need are a set of unique node embeddings + the graph-specific expressive decoder (no need for graph-aware message passing). Do the authors have any reason why the decoder needs to use the graph structure (as a result, it can only predict L-hop node proximity)?
> > > > > > > > >
> > > > > > > > > ===
> > > > > > > > > I updated my score from 1 to 3 to reflect the authors' effort in implementing and evaluating the simple method. Still, I am inclined not to accept this paper since I find the technical contribution to be minimal compared to the P-GNN paper.

---

> > > > > > > > > > ### Author Response · Authors · 2020-11-21
> > > > > > > > > > **Thanks for Your Comments and Improving the Score**
> > > > > > > > > >
> > > > > > > > > > We thank the reviewer for summarizing the comments and improving the score. Here are our responses.
> > > > > > > > > >
> > > > > > > > > > Q: Results on node classification
> > > > > > > > > > A: We greatly outperform P-GNN on node classification (since the reviewer seems to think this baseline is most relevant) and retain comparable results with other GNNs, thus making our proposed method a flexible solution to both proximity-aware and permutation-equivariant tasks.
> > > > > > > > > >
> > > > > > > > > > Q: The method is the baseline of P-GNN
> > > > > > > > > > A: As explained in previous comments, our method is **different** from simply using one-hot IDs (using one-hot IDs + using Gaussian to initialize + fixing the first-layer parameters is only similar to **the stochastic part** of our proposed method). Otherwise, we could not achieve better results than P-GNN.
> > > > > > > > > >
> > > > > > > > > > Q: Theoretical claims hinge on unrealistic assumptions
> > > > > > > > > > A: The assumptions mostly inherit from previous work [1] (bijective functions, countable feature space). Besides, as explained in previous comments, we have a more practical version, i.e., the linear variant of SMP and Theorem 2. Experimental results also support that the linear variant is expressive enough in most cases.
> > > > > > > > > > [1] How Powerful are Graph Neural Networks, ICLR 2019
> > > > > > > > > >
> > > > > > > > > > Q: Although the authors do use the graph structure in their proof of Theorem 3, in a transductive setting, it still seems that all we need are a set of unique node embeddings + the graph-specific expressive decoder (no need for graph-aware message passing). Do the authors have any reason why the decoder needs to use the graph structure (as a result, it can only predict L-hop node proximity)?
> > > > > > > > > > A: We have explained multiple times that we **do not** use the graph structure in the decoder, i.e., we only use the embeddings obtained by GNNs $H$, the stochastic matrix $E$, and the similarity function $\mathcal{S}(\cdot)$. Please let us know which step of the proof you feel we have used the graph structure. This also helps to answer your questions: since only L-hop proximities are encoded in $H$, we can only preserve them.

---

> > > > > > > > > > > ### Comment · AnonReviewer3 · 2020-11-21
> > > > > > > > > > > **Clarification**
> > > > > > > > > > >
> > > > > > > > > > > Re: We have explained multiple times that we do not use the graph structure in the decoder.
> > > > > > > > > > >
> > > > > > > > > > > Yes, I understand your method does not. My point is that in the transductive setting, the decoder **can** use the graph structure, and there is no need to perform message passing on the encoder side. The randomly-sampled Gaussian vectors can be directly used, and the expressive decoder can memorize the proximity between all the pairs of nodes in the graph. Is there any issue with this approach? (This approach is kind of similar to node2vec or conventional matrix factorization techniques.)

---

> > > > > > > > > > > > ### Author Response · Authors · 2020-11-21
> > > > > > > > > > > > **Responses**
> > > > > > > > > > > >
> > > > > > > > > > > > We thank the reviewer for clarifying the question (there may be typos in the previous question).
> > > > > > > > > > > >
> > > > > > > > > > > > We think not adopting graph structures in the decoder is the standard setting of using GNNs and node embeddings--otherwise, you can build a do-nothing GNN and name the combination of a real GNN and decoder as a new "decoder", which is essentially not in accordance with the encoder-decoder framework, i.e., encoding structral information in the encoder (i.e., GNNs) and decoding information from obtained node representations. Please see Figure 3 of survey [1] or Figure 4.2 of book [2] for the encoder-decoder setting.  Notice that even for network embedding methods such as node2vec or MF, the graph structure is not utilized after obtaining the node embedding vectors.
> > > > > > > > > > > > [1] Representation Learning on Graphs: Methods and Applications, Bulletin of the IEEE 2017.
> > > > > > > > > > > > [2] Deep Learning on Graphs, Cambridge University Press, 2020.

---

### Official Review · AnonReviewer2 · 2020-10-27
**A well-rounded analysis of a simple method.**

**Rating:** 7
**Confidence:** 4

**Review:**

The authors propose to add random node features to the input of message passing graph neural networks for them to become proximity-aware. The paper provides exhaustive theoretical and empirical analysis of this idea, highlighting the advantageous properties.

Strengths:
- The paper is well written.
- The authors substantiate their claims with proofs and experiments.
- The method and proofs seem to be mathematically sound.
- Including the experiments in the appendix, the empirical analysis is very exhaustive and seems to be reproducible.
- The method is simple but effective, which is good. It also does not lead to a significant computational overhead but fits nicely into the existing message passing framework with linear time complexity (in number of edges).
- The paper provides a formal framework and analysis for a trick that has already been used successfully in practice.

Comments and Questions:
- The work of Sato et al. [1] seems to be closer to this work than the authors let know. I would welcome a more in-depth discussion than given in related work, even if the paper is only on arxiv.
- Definition 4 should go to the main text, as it is crucial for understanding.
- The proof of Theorem 1 shows the result only for graphs with 2 connected components. I think the theorem also holds for connected graphs (with the right automorphisms), which is important. If it wouldn't, the result would not always be relevant for practice. I think the current way of proving the theorem lacks that insight and discussion. Isn't it possible to prove the theorem for all graphs with a certain automorphism, regardless of number of connected components?
- Can the authors verify the suspicion that the non-linear variants overfit (line 270) by presenting the results on training data?
- It might be of interest to the authors that the idea was already used in a practical graph matching method by [2] (page 5, last paragraph), although without any theoretical analysis. It is nice to have a formal framework that justifies the application of this trick.
- In general, GNNs to solve matching tasks are a very fitting application for the proposed method, which the authors do not consider. They often rely on comparing distance measures in both domains, thus need proximity awareness.

Typos:
- Line 87: "computationally expansive" -> expensive
- Line 607: "computationally expansive" -> expensive

All in all, there is not much to complain about. The paper achieves what it sets out to do in providing an exhaustive theoretical and empirical analysis of a simple but effective idea. The method itself is straight-forward and also not entirely novel. However, the formal framework and analysis is a contribution that is of interest to the community. Due to the shown properties and the high efficiency of the approach, the paper can have significant impact on future GNN architectures in practice. I therefore recommend to accept the paper.

[1] Sato et al., Random features strengthen graph neural networks. arXiv:2002.03155

[2] Fey et al., Deep Graph Matching Consensus, ICLR 2020

---

> ### Author Response · Authors · 2020-11-18
> **To Reviewer 2**
>
> Thanks for your kind words and detailed comments. Here are our responses to your comments.
>
> Q1. The work of Sato et al. [1] seems to be closer to this work than the authors let know. I would welcome a more in-depth discussion than given in related work, even if the paper is only on arxiv.
> [1] Sato et al., Random features strengthen graph neural networks. arXiv:2002.03155
> A1. Thanks for your suggestion. We have revised the related works as follows: ”For example, Sato et al. (Sato et al., 2020) novelly show that random numbers can enhance GNNs in tackling two important graph-based NP problems with a theoretical guarantee, namely the minimum dominating set and the maximum matching problem…Our work differs in that we systematically study how to preserve permutation-equivariance and proximity-awareness simultaneously in a simple yet effective framework, which is a new topic different from these existing works. Besides, we theoretically prove that our proposed method can preserve walk-based proximities by using the random projection literature. We also demonstrate the effectiveness of our method on various large-scale benchmarks for both node- and edge-level tasks, while no similar results are reported in the literature.”
>
> Q2. Definition 4 should go to the main text, as it is crucial for understanding.
> A2: Thanks for the suggestion. We have moved the definition to the main paper in the revised version.
>
> Q3. The proof of Theorem 1 shows the result only for graphs with 2 connected components. I think the theorem also holds for connected graphs (with the right automorphisms), which is important. If it wouldn't, the result would not always be relevant for practice. I think the current way of proving the theorem lacks that insight and discussion. Isn't it possible to prove the theorem for all graphs with a certain automorphism, regardless of number of connected components?
> A3. Thanks for this insightful and constructive comment. We indeed find that we can prove the theorem using one connected component under a mild assumption (i.e., the walk-based proximity is of finite length). We have provided this additional proof in Appendix A.1 in the revised version. The essential idea is to construct automorphism using three copies of the graph and three bridges to connect these copies. Please refer to the revised manuscript for details.
>
> Q4. Can the authors verify the suspicion that the non-linear variants overfit (line 270) by presenting the results on training data?
> A4: Thank you for this insightful comment. We have provided your suggested experiments in Appendix B.6 in the revised version. The results show that non-linear variants of GNNs such as GAT indeed exhibit more serious overfitting, i.e., the margins between the training accuracies and the testing accuracies are usually larger than the linear variant SGC. Besides, non-linear variants are also difficult to train, i.e., the training accuracies are sometimes also worse than the linear variant SGC. We have clarified our expressions as follows: “Some plausible reasons include that the additional model complexity brought by non-linear operators makes the models tend to overfit and also difficult to train.”
>
> Q5. It might be of interest to the authors that the idea was already used in a practical graph matching method by [2] (page 5, last paragraph), although without any theoretical analysis. It is nice to have a formal framework that justifies the application of this trick.
> [2] Fey et al., Deep Graph Matching Consensus, ICLR 2020
> A5. Thanks for your notice on this interesting relevant literature. We have added it to the related works in the revised version.
>
> Q6. In general, GNNs to solve matching tasks are a very fitting application for the proposed method, which the authors do not consider. They often rely on comparing distance measures in both domains, thus need proximity awareness.
> A6. Thanks for your constructive suggestion. We will consider trying our proposed method in graph matching tasks.
>
> Q7: Typos:
> Line 87: "computationally expansive" -> expensive
> Line 607: "computationally expansive" -> expensive
> A7: Thanks for pointing them out. We have fixed them in the revised version.
>
> Q8. All in all, there is not much to complain about. The paper achieves what it sets out to do in providing an exhaustive theoretical and empirical analysis of a simple but effective idea. The method itself is straight-forward and also not entirely novel. However, the formal framework and analysis is a contribution that is of interest to the community. Due to the shown properties and the high efficiency of the approach, the paper can have significant impact on future GNN architectures in practice. I therefore recommend to accept the paper.
> A8: Thank you for your recognition that our paper can significantly impact future GNNs! We intend to further explore this direction in the near future.

---

> > ### Comment · AnonReviewer2 · 2020-11-20
> > **Thank you for the response.**
> >
> > I thank the authors for the additional effort, polishing the work further. My evaluation remains the same.
> >
> > After reading the other reviews, discussions and revisions carefully, I'd like to explicitly mention that I do not share the opinion of Reviewer3 that this work contains only trivial results.
> >
> > The method itself certainly is simple. It does not really matter if you present it as noise as input or as a non-trainable, randomly initialized layer following one-hot features, as both variants are similarly straight-forward and seem to be equivalent. I do not consider the fact that those views are interchangeable as argument for any position.
> >
> > In general, I think the complexity of a presented method should not be a strong deciding factor. A lot of past high-impact work presents, analyzes and evaluates simple methods that can be easily adapted and are efficient. Both is the case in this work. As a researcher more focused on practical applications, I might even favor simple, effective approaches over complex ones that have strong theoretical properties but are inapplicable in practice. The approach presented in this paper has practical advantages over the previous work (P-GNNs).
> >
> > I agree that the approach is not entirely novel since it has been applied before as an optimization trick in some works. However, to my knowledge, there is no work which has this idea as a main contribution and makes an effort analyzing it's advantages, and where those advantages come from. This, in my opinion, is a non-trivial and potentially impactful contribution. Future GNN research can build upon this work, which previous work might not allow due to practical limitations.

---

### Official Review · AnonReviewer1 · 2020-10-28
**Review...**

**Rating:** 6
**Confidence:** 2

**Review:**

This paper proposed a proximity-aware graph neural network while maintaining the permutation equivariance property. The proposed model, dubbed as stochastic message passing (SMP), arguments the existing GNNs with stochastic node representations. The author proved the proposed method can model proximity-aware representations based on random projection theory. The experimental results show that the SMP can be used for multiple graphs and tasks.

Although the proposed technique is relatively simple, the theorem shows that why such stochastic representations are beneficial for proximity-aware tasks. The experimental results also suggest that this simple technique is quite effective in many tasks. There are a few things that I'd like to comment on or ask listed below:

- Multiple tasks have been conducted to show the performance of the proposed model, but it is unclear which dataset or which task requires to be proximity-aware. It would be great if there's some quantitative metric that shows the importance of proximity in a given task. For example, the number of automorphic node pairs (within k-hop) and the ratio of label (dis)agreement would be a possible metric in node classification tasks. This will characterize the differences between datasets and highlight the contribution of the proposed method. Additionally, showing some examples of automorphic node pairs and the performance on these nodes could demonstrate the difference between models.

- Resampling random matrix at each epoch is emphasized multiple times in the manuscript, but without any empirical experiments. Would it be beneficial to resample this random matrix at every epoch? Although in theory, it would be possible to learn GNN that preserves node proximity, if a given task doesn't need to model proximity-aware representations, random resampling may hinder the convergence of the proposed method.

- There was a bug in the official release of the P-GNN paper, which has been recently fixed (please check the GitHub pull request history at https://github.com/JiaxuanYou/P-GNN/pull/12). I wonder which codebase the authors used for the experiments. / It would be also good if there's some comment on what makes P-GNN memory hunger. Is it because of improper implementation or because of some inherent limitations?

---

> ### Author Response · Authors · 2020-11-18
> **To Reviewer 1**
>
> Thanks for your recognition and comments. Here are our responses to your questions.
>
> Q1: Multiple tasks have been conducted to show the performance of the proposed model, but it is unclear which dataset or which task requires to be proximity-aware. It would be great if there's some quantitative metric that shows the importance of proximity in a given task. For example, the number of automorphic node pairs (within k-hop) and the ratio of label (dis)agreement would be a possible metric in node classification tasks. This will characterize the differences between datasets and highlight the contribution of the proposed method. Additionally, showing some examples of automorphic node pairs and the performance on these nodes could demonstrate the difference between models.
> A1: Thank you for this insightful suggestion. We agree that a metric to quantify to what degree a task is proximity-aware or permutation-equivariant will be helpful for further studying GNNs. However, we find that such a metric is difficult to design due to two reasons. Firstly, only comparing whether two nodes are strictly automorphic, as you have suggested, is important but may not be sufficient since if the local structures of two nodes are very similar (but not strictly automorphic), the results of GNNs will also be affected. Thus, we need to quantify the similarities between local structures of different nodes (rather than only comparing whether they are exactly automorphic). However, this is an interesting research question itself, e.g., commonly studied in graph kernels. Secondly, besides this metric related to permutation-equivariance, we need another metric to quantify whether the task can benefit from proximity-awareness, which is also non-trivial to develop. All things considered, we leave studying this interesting topic as important future works.
>
> Q2: Resampling random matrix at each epoch is emphasized multiple times in the manuscript, but without any empirical experiments. Would it be beneficial to resample this random matrix at every epoch? Although in theory, it would be possible to learn GNN that preserves node proximity, if a given task doesn't need to model proximity-aware representations, random resampling may hinder the convergence of the proposed method.
> A2: Thank you for this comment. As we explain in Section 4.1, we find that fixing the stochastic matrix can help the model to memorize the stochastic representation and distinguish different nodes, while resampling the matrix in each epoch will make our method more capable of handling nodes not seen during training. As a result, we fix the stochastic matrix on transductive datasets and resample the stochastic matrix on inductive datasets. Following your comments, we have added an ablation study in Table 12 in Appendix B.5 in the revised manuscript. The results verify that our design leads to better results in most cases, i.e., fixing the stochastic matrix on transductive datasets and resampling the stochastic matrix on inductive datasets. Please refer to the revised manuscript for details.
>
> Q3: There was a bug in the official release of the P-GNN paper, which has been recently fixed (please check the GitHub pull request history at https://github.com/JiaxuanYou/P-GNN/pull/12). I wonder which codebase the authors used for the experiments. / It would be also good if there's some comment on what makes P-GNN memory hunger. Is it because of improper implementation or because of some inherent limitations?
> A3: Thank you for this notice. We indeed adopt the implementation by the authors. Since this bug is fixed after our submission, the results in the paper are based on the implementation before the fix. We have tried the fixed codes and find the results comparable or even slightly worse than the results reported in the paper. As for the memory issue, we think it is caused by the mechanism of P-GNN, i.e., explicit construction of the anchor nodes and performing message-passing on them.

---

### Official Review · AnonReviewer4 · 2020-11-01
**A significant improvement over P-GNN**

**Rating:** 8
**Confidence:** 4

**Review:**

Since the publication of P-GNN (You et al., 2019), it has become clear to the graph ML community that node positional information can be effectively leveraged for link prediction and pairwise node classification tasks.
This paper introduces SMP, a novel stochastic message passing approach that preserves both permutation-equivariance (common to GNN models) and node proximities. Extensive experimental results show that SMP not only achieves competitive performance on many common graph ML datasets, but also succeeds to combine together the expressiveness of a standard GNN with P-GNN  (without incurring the scalability problem of P-GNN).

I thoroughly enjoyed reading this paper, both for the insights and the technical soundness. I have very few remarks about the paper, as I believe that i) SMP is an ingenious idea, ii) the experimental setting is adequate, iii) the quality of the writeup is high, iv) and the results appear to be reproducible.

If I had to nitpick, I'd say that part of Table 7 (currently in the Appendix) belongs to the main paper, as I was convinced about the superior runtime performance of SMP only after reading those numbers. If the avg running time for an SMP epoch was significantly larger than the one for GAT (for instance), I would have considered SMP yet another specialized model.
Instead, given that the GPU consumption (both in terms of computation and memory) is similar to any other GNN model, I believe SMP could be adopted as a more flexible graph ML method (which would avoid having to choose a method given the target task, e.g., node classification vs. link prediction).

One remark about L278: SMP cannot be considered anymore SotA for ogbl-ppa. The current SotA is more than 10 points above the performance achieved by SMP.

---

> ### Author Response · Authors · 2020-11-18
> **To Reviewer 4**
>
> Thank you for your recognition and kind words! Here are our responses to your comments.
>
> Q: If I had to nitpick, I'd say that part of Table 7 (currently in the Appendix) belongs to the main paper, as I was convinced about the superior runtime performance of SMP only after reading those numbers. If the avg running time for an SMP epoch was significantly larger than the one for GAT (for instance), I would have considered SMP yet another specialized model. Instead, given that the GPU consumption (both in terms of computation and memory) is similar to any other GNN model, I believe SMP could be adopted as a more flexible graph ML method (which would avoid having to choose a method given the target task, e.g., node classification vs. link prediction).
> A: Thanks for the suggestion. We have moved the table to the main paper in the revised version. We also appreciate the reviewer’s comments that the high-efficiency makes our proposed method a flexible approach to handle graph-based machine learning tasks.
>
> Q: One remark about L278: SMP cannot be considered anymore SotA for ogbl-ppa. The current SotA is more than 10 points above the performance achieved by SMP.
> A: Thanks for the notice that SEAL recently achieves a new SOTA on PPA (which happens after our submission). We have clarified our expression in the revised version. Besides, since SEAL is a GNN variant specifically designed for link prediction, we are interested to see whether our proposed method can further improve SEAL.

---

### Author Response · Authors · 2020-11-18
**Summary of Changes**

We would like to thank all the reviewers again for their thoughtful comments on our paper. Based on their comments, we have made the following changes, which further improve the quality of our paper.

* We have moved Definition 4 (preserve walk-based proximity, lines 142-149 in the revised manuscript) and Table 5 (running time comparison, Page 9 in the revised manuscript) into the main paper from the appendix since they are important to our proposed method.
* We have improved related works (lines 88-101 in the revised manuscript) by revising the comparison with recent works and adding one related work (Fey et al., 2020).
* We have added one ablation study (Table 12, Appendix B.5 in the revised manuscript) to compare whether the stochastic matrix E is fixed or resampled during different training epochs.
* We have given an alternate proof of Theorem 1 (lines 526-550, Appendix A.1 in the revised manuscript) by maintaining one connected component in constructing the counter-examples.
* We have added one subsection (Appendix B.6 in the revised manuscript) to investigate how linear and non-linear variants of permutation-equivariant GNN perform for the link prediction task.
* We have slightly modified Definition 4 and Theorem 3 to prevent misunderstanding and corner cases (do not affect other parts of the paper).
* We have clarified some expressions and fixed some typos.

We would be happy to address any other questions or comments.

---

### Decision · Program_Chairs · 2021-01-07
**Final Decision**

**Decision:**

Reject

**Comment:**

This work proposes a modification of a GNN architecture by feeding random node features to bootstrap the message propagation. This enables the discriminability of automorphic node pairs with a lightweight, simple change. Experiments are reported showing improvements over baselines.
Reviewers had mixed impressions of this work. On one hand, they found the proposed model principled and with strong empirical performance. On the other hand, they perceived a general lack of novelty and a somewhat misleading theoretical analysis. After careful review, the AC ultimately believes that this work does require an extra iteration that further solidifies the contributions and aligns the theoretical analysis with the empirical performance. In particular, the use of random initialization is folklore in the GNN literature, especially with regards to spectral methods (e.g. power iterations are typically initialized using a random vector, and these constitute the simplest forms of linear GNNs). The authors are encouraged to address these comparisons with further detail, as well as the excellent feedback given by the reviewers.